# Glucose restriction drives spatial reorganization of mevalonate metabolism

Sean Rogers[1], Hanaa Hariri[1,2], N Ezgi Wood[1], Natalie Ortiz Speer[1], W Mike Henne[1]*

[1]Department of Cell Biology, University of Texas Southwestern Medical Center, Dallas, United States; [2]Department of Biological Sciences, Wayne State University, Detroit, United States

**Abstract** Eukaryotes compartmentalize metabolic pathways into sub-cellular domains, but the role of inter-organelle contacts in organizing metabolic reactions remains poorly understood. Here, we show that in response to acute glucose restriction (AGR) yeast undergo metabolic remodeling of their mevalonate pathway that is spatially coordinated at nucleus-vacuole junctions (NVJs). The NVJ serves as a metabolic platform by selectively retaining HMG-CoA Reductases (HMGCRs), driving mevalonate pathway flux in an Upc2-dependent manner. Both spatial retention of HMGCRs and increased mevalonate pathway flux during AGR is dependent on NVJ tether Nvj1. Furthermore, we demonstrate that HMGCRs associate into high-molecular-weight assemblies during AGR in an Nvj1-dependent manner. Loss of Nvj1-mediated HMGCR partitioning can be bypassed by artificially multimerizing HMGCRs, indicating NVJ compartmentalization enhances mevalonate pathway flux by promoting the association of HMGCRs in high molecular weight assemblies. Loss of HMGCR compartmentalization perturbs yeast growth following glucose starvation, indicating it promotes adaptive metabolic remodeling. Collectively, we propose a non-canonical mechanism regulating mevalonate metabolism via the spatial compartmentalization of rate-limiting HMGCR enzymes at an inter-organelle contact site.

*For correspondence:
mike.henne@utsouthwestern.edu

**Competing interests:** The authors declare that no competing interests exist.

## Introduction

The complexity of eukaryotic metabolism requires spatial organization, so membrane-bound and membraneless organelles compartmentalize enzymes into distinct sub-cellular regions. Enzyme partitioning into spatially defined domains is a cellular organizational principle observed in cytoplasmic assemblies (*Chan et al., 2015*), enzyme homo-polymers (*Meredith and Lane, 1978*) and biomolecular condensates (*Banani et al., 2017*; *Jin et al., 2017*). Such enzymatic assemblies, or metabolons, enhance or fine-tune metabolic flux, locally sequester toxic intermediates, or limit product shunting to competing pathways. Enzyme partitioning has also been exploited in synthetic bioengineering to artificially force enzymes into close proximity to drive local reactions or metabolic channeling through enzymatic cascades (*Dueber et al., 2009*).

Inter-organelle contact sites also serve as platforms for enzymatic organization and lipid synthesis reactions. In particular, the endoplasmic reticulum (ER) partitions lipogenic processes like non-vesicular lipid transport and organelle biogenesis at inter-organelle junctions (*Lev, 2012*; *Hariri et al., 2018*; *Friedman et al., 2011*). In yeast and humans, mitochondrial-associated membranes with the ER (MAMs) harbor enhanced metabolic activity to support reactions such as the biosynthesis and transport of phosphatidylserine (*Vance, 1990*; *Gaigg et al., 1995*). More recent work has highlighted the role of ER-mitochondrial contacts defining distinct sub-domains within mitochondria that support the formation of multi-enzyme complexes that drive Coenzyme Q synthesis (*Subramanian et al., 2019*; *Eisenberg-Bord et al., 2019*). In yeast, the nuclear envelope (continuous with the ER network) also makes extensive contact with the vacuole (equivalent to the lysosome),

forming a nucleus-vacuole junction (NVJ). Formed through hetero-dimerization of nuclear envelope protein Nvj1 and vacuole-associated protein Vac8, the NVJ is a disc-shaped inter-organelle contact that acts as a multi-functional platform to organize lipid transport, fatty acid synthesis, and lipid droplet (LD) biogenesis, particularly during metabolic stress (*Hariri et al., 2018*; *Murley et al., 2015*; *Kvam et al., 2005*). Despite these insights, few studies mechanistically dissect how metabolic cues regulate enzyme recruitment and compartmentalization at inter-organelle contacts. Furthermore, how inter-organelle junctions enable enzyme spatial organization to fine-tune or enhance metabolic flux remains poorly described. Such inter-organelle crosstalk is particularly pertinent to understanding adaptive responses to stresses such as glucose starvation, which is characterized by drastic decreases in cellular ATP as well as changes in cytoplasmic pH and fluidity that alter macromolecular trafficking (*Joyner et al., 2016*; *Munder et al., 2016*).

Here, we use budding yeast as a genetically enabling model system to dissect the role of ER-lysosome contacts (e.g. the NVJ) as organizational platforms in mevalonate metabolism. We find that in response to acute glucose restriction (AGR), yeast actively partition and selectively retain HMG-CoA Reductase (HMGCR) enzymes at the NVJ in a Nvj1 and Upc2-dependent manner. This enzyme sub-compartmentalization enhances mevalonate pathway flux and promotes mevalonate and ultimately sterol-ester biosynthesis during AGR-induced metabolic remodeling. Furthermore, we find loss of HMGCR NVJ partitioning impacts metabolic remodeling following glucose starvation; failure to compartmentalize HMGCRs affects resumption of yeast growth following re-introduction to glucose containing media, but can be rescued by addition of exogenous mevalonate. We also find that NVJ compartmentalization of HMGCRs is accompanied by an Nvj1-dependent association of HMGCRs into high-molecular-weight complexes. Interestingly, both the growth resumption delay and mevalonate production phenotypes observed in Nvj1 knock-out cells can be rescued by the artificial multimerization of HMGCRs via a tetramerizing tag.

## Results

### Yeast Hmg-CoA reductases (HMGCRs) inducibly partition at the nucleus-vacuole junction (NVJ) in response to acute glucose restriction (AGR)

Given that ER-mediated contact sites in *Saccharomyces cerevisiae* act as lipogenic domains, we visually screened GFP-tagged neutral lipid metabolism enzymes for signs of compartmentalization at ER inter-organelle contact sites in yeast exposed to AGR. Note that AGR is operationally defined here as culturing yeast in synthetic complete (SC) media containing 2% glucose, then briefly centrifuging them and placing them into SC media containing 0.001% glucose. Screening revealed that endogenously GFP-tagged HMG-CoA Reductases (HMGCR) Hmg1 and Hmg2, which localize throughout ER network and particularly on the nuclear envelope (NE), enriched at the regions where the nuclear surface was in contact with the vacuole following AGR (*Figure 1A*). This nucleus-vacuole interface was confirmed to be the NVJ as Hmg1-GFP co-localized with NVJ tether Nvj1-mRuby3 following AGR treatment for 4 hr (*Figure 1B*). As Hmg1 and Hmg2 are functionally redundant in mevalonate synthesis, a central metabolite that supplies diverse cellular pathways including sterol biogenesis, we chose to dissect the mechanisms underlying Hmg1 NVJ partitioning. To investigate whether the NVJ was required for Hmg1 partitioning, we imaged Hmg1-GFP in the absence of NVJ tethers Nvj1 and Vac8 (*Pan et al., 2000*), and found they were required for Hmg1-GFP partitioning during AGR (*Figure 1C*). Furthermore, time-lapse imaging revealed that Hmg1-GFP NVJ partitioning peaked after ~4 hr of introducing yeast to AGR (*Figure 1D–F*). During this period, Hmg1-GFP was ~15 times enriched at the NVJ relative to other NE regions. Collectively, this suggests that AGR induces the time-dependent compartmentalization of Hmg1/2 at the NVJ in a Nvj1 and Vac8 dependent manner.

### NVJ compartmentalization of HMGCRs is independent of other mevalonate pathway enzymes and lipid trafficking proteins Osh1 and Ltc1

Inter-organelle contacts are implicated as domains coordinating the recruitment of supra-molecular enzyme complexes that constitute metabolic pathways. The HMGCRs are the rate-limiting enzymes

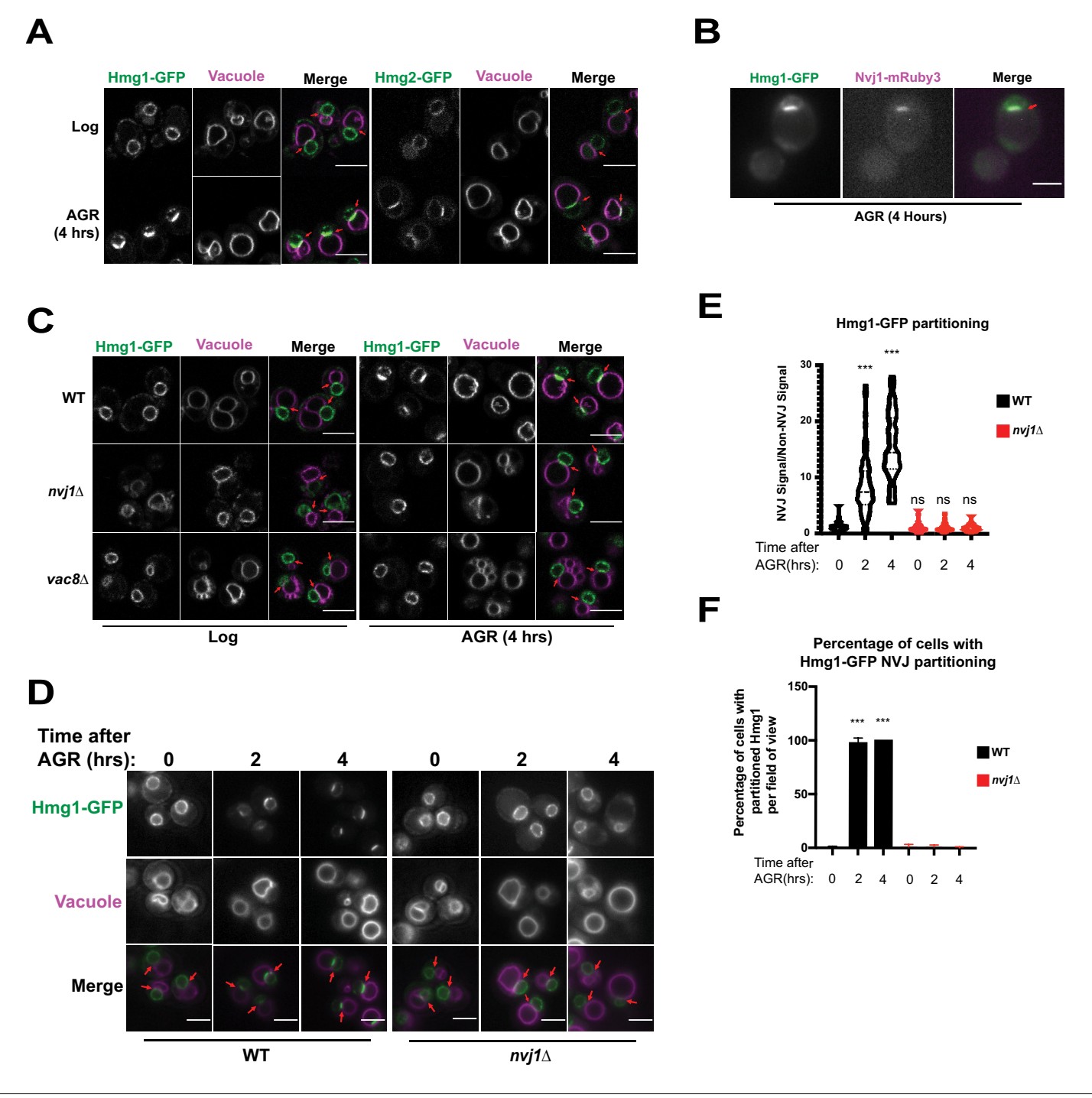

**Figure 1.** The HMG-CoA Reductases, Hmg1 and Hmg2, enrich at the NVJ in an Nvj1-dependent manner during AGR. (**A**) Confocal microscopy images of yeast expressing endogenously tagged Hmg1-GFP or Hmg2-GFP (green) grown in SC media with 2% glucose to an OD of 0.5 (Log), or in 0.001% glucose for 4 hr (AGR). Vacuoles (magenta) were visualized by staining with FM4-64. Red arrows represent the relative location of the Nucleus-Vacuole Junction (NVJ), or where the nuclear envelope is closely apposed to the vacuole in *nvj1Δ* cells. Scale bars = 5 μm. (**B**) Epifluorescence microscopy images showing the overlap of Hmg1-GFP with Nvj1-mRuby3 after cells were exposed to four hours of AGR. Red arrows represent the location of the NVJ. Scale bars = 5 μm (**C**) Confocal microscopy images of yeast expressing endogenously tagged Hmg1-GFP (green) in wild type (WT), *nvj1Δ*, or *vac8Δ* yeast. Red arrows represent the location of NVJ. (**D**) Epifluorescence microscopy images of WT or *nvj1Δ* yeast expressing endogenously tagged Hmg1-GFP growing in 2% glucose to an OD of 0.5 (Log) or in 0.001% glucose for 2 or 4 hr (AGR). (**E**) Quantification of Hmg1 partitioning measured by line scan across the nuclear envelope (NE) and plotted as the ratio of Hmg1-GFP intensity at the NVJ relative to Hmg1-GFP intensity opposite the NVJ. Scale bars represent 5 μm. (Brown-Forsyth and Welch ANOVA. N > 51 cells. ***p-value<0.001). (**F**) Quantification of percentage of cells displaying

*Figure 1 continued on next page*

Figure 1 continued

Hmg1 NVJ partitioning per field of view. Any cell with partitioning >2.0 was considered as displaying Hmg1 NVJ partitioning. (Brown-Forsyth and Welch ANOVA. N > 51 cells. ***p-value<0.001).

of the mevalonate pathway that generates ergosterol, the cholesterol analog of yeast. To dissect whether other proteins along the mevalonate pathway also partition at the NVJ during AGR, we imaged 20 endogenously mNeonGreen (mNG)-tagged proteins involved in ergosterol biosynthesis under ambient and AGR stress, but surprisingly none detectably enriched at the NVJ (*Figure 2A*). As the NVJ has also been proposed to function in sterol transport between the vacuole and ER network, we also examined whether loss of the NVJ-resident proteins Osh1 and Lam6/Ltc1, implicated in the inter-organelle trafficking of lipids and/or sterols, would impact Hmg1-GFP NVJ recruitment (*Murley et al., 2015*; *Kvam and Goldfarb, 2004*). Hmg1-GFP efficiently partitioned at the NVJ following AGR in both *osh1Δ* and *lam6/ltc1Δ* yeast (*Figure 2B*). Collectively, this suggests that detectable Hmg1/2 NVJ enrichment is unique among enzymes involved in ergosterol metabolism, and is therefore unlikely to constitute a 'classical' multi-enzyme metabolon.

## Hmg1-GFP is selectively retained at the NVJ in a Nvj1-dependent manner

Next, we investigated the mechanistic basis for NVJ compartmentalization of Hmg1-GFP. We began by interrogating whether the NVJ-partitioned and non-partitioned Hmg1-GFP pools were physically segregated along the NE. Using fluorescence recovery after photobleaching (FRAP), we determined that the non-partitioned Hmg1-GFP pool can enter the NVJ (*Figure 3A,B*); therefore, there is likely no diffusion barrier at the NVJ allowing selective entry of only specific pools of Hmg1-GFP protein. Next we interrogated whether Hmg1-GFP was selectively retained after entry into the NVJ region. Fluorescence loss in photobleaching (FLIP) revealed the average lifetime of NVJ-partitioned Hmg1-GFP was >100 s, compared to only ~25 s for Hmg1-GFP along the rest of the NE (*Figure 3C,D*). Given that most of the Hmg1-GFP signal within the NVJ-partitioned population was not photobleached to completion, calculated halftimes may underestimate the actual halftime. Collectively, this indicates that upon AGR, Hmg1-GFP is recruited and selectively retained at the NVJ. Surprisingly, the halftime of Hmg1-GFP was also significantly increased during AGR in *nvj1Δ* cells. This may be connected to a change in cytoplasmic viscosity which has been examined previously (*Joyner et al., 2016*); however, further analysis is required to determine if this reduced mobility is specific to Hmg1-GFP or a general consequence of AGR.

Next, we dissected which regions of Hmg1 and Nvj1 were required for Hmg1-GFP NVJ retention. We generated truncated versions of GFP-tagged Hmg1 containing only its integral transmembrane (TM) region and lacking its cytoplasm-exposed catalytic domain. This Hmg1$_{1-525}$-GFP truncation was sufficient to localize to the NVJ during AGR, indicating the integral membrane region of Hmg1 was sufficient for NVJ recruitment (*Figure 3E,F*). To dissect the Nvj1 regions required for Hmg1-GFP recruitment, we expressed mNg-tagged Nvj1 fragments in yeast co-expressing Hmg1-mRuby3 (*Figure 3G*). A Nvj1-mNG chimera lacking its cytoplasm-exposed region and containing a vacuole-binding PX domain in place of its Vac8-binding domain (Nvj1-PX) was sufficient to tether between the nucleus and vacuole, and was sufficient to partition Hmg1-mRuby3 at the NVJ during AGR, indicating the Nvj1 cytoplasmic domain is not required for Hmg1 recruitment (*Figure 3—figure supplement 1A*). Similarly, a Nvj1-mNG chimera with its TM region replaced with the TM of Nvj2 (Nvj1$_{Nvj2TM}$) also partitioned Hmg1-mRuby3 at the NVJ, indicating the TM and cytoplasmic regions are not required for Hmg1-mRuby3 recruitment (*Figure 3—figure supplement 1A*). However, expression of a full length Nvj1-mNG lacking residues 15–30 (Nvj1$_{Δ15-30}$) of its luminal region formed an NVJ contact but failed to recruit Hmg1-mRuby3. Re-addition of six residues to this construct (Nvj1$_{Δ15-24}$) rescued Hmg1-mRuby3 recruitment. Furthermore, mutation of arginine and lysine residues at positions 28/29 to alanines (Nvj1$_{RK→AA}$) also ablated AGR-induced recruitment of Hmg1 to the NVJ, suggesting residues 25–30 of the Nvj1 luminal domain play a key role in Hmg1-mRuby3 NVJ partitioning (*Figure 3G–I*). Collectively, these data indicate that the ER embedded region of Hmg1 is recruited to the NVJ in a manner requiring the ER-luminal Nvj1 region, and that Hmg1

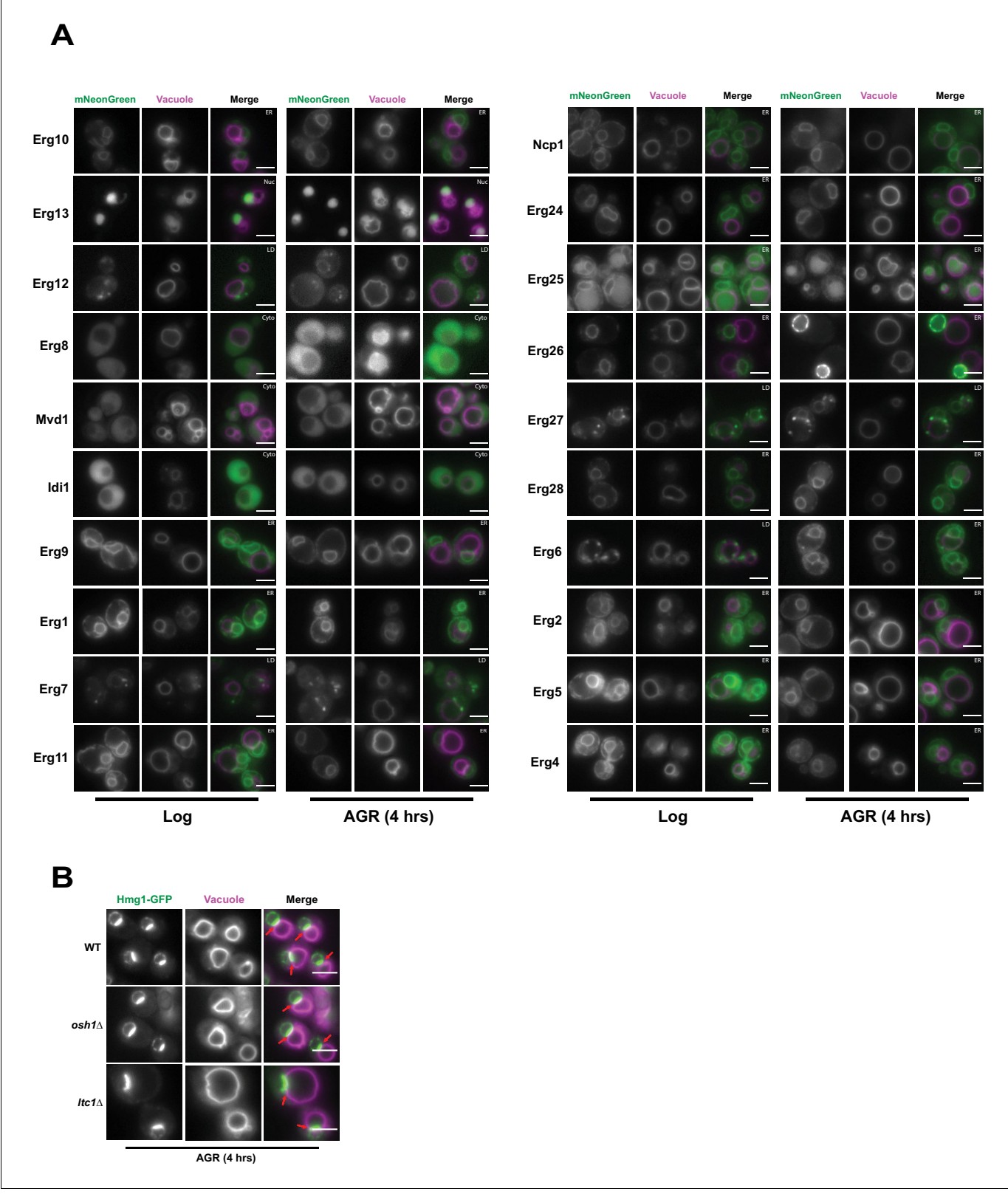

**Figure 2.** Hmg1 NVJ partitioning is specific amongst ergosterol biosynthesis proteins and independent of NVJ-associated sterol transport proteins. (**A**) Epifluorescence microscopy images of twenty ergosterol biosynthesis proteins endogenously tagged with mNeonGreen (mNg) imaged in log phase and after 4 hr of AGR. Cells were co-stained with FM4-64 (magenta) to visualize vacuoles. Cellular compartments occupied by each protein are

*Figure 2 continued on next page*

*Figure 2 continued*

designated in white text in the top right corner of each image. Scale bars = 5 µm. (**B**) Epifluorescence microscopy images of cells endogenously tagged with Hmg1-GFP in wild type (WT), *osh1Δ*, or *ltc1Δ* cells exposed to 4 hr of AGR. Scale bars = 5 µm.

partitioning is not dependent on the Hmg1 cytoplasmic catalytic domain nor the Nvj1 cytoplasmic/vacuole binding region.

## NVJ partitioning of Hmg1 is Upc2 dependent and correlates with sterol-ester biosynthesis

The NVJ was previously identified as a site for starvation-induced microautophagy of the nucleus (*Roberts et al., 2003*), where NVJ-associated proteins accumulate and are subsequently engulfed by the vacuole. HMGCRs are also known to undergo proteasomal degradation during specific metabolic cues. To determine if NVJ partitioning of Hmg1-GFP regulated Hmg1 protein abundance or turnover, we performed immunoblot analysis of Hmg1-GFP in log-phase and 4 hr AGR-treated cells. AGR increased steady-state Hmg1-GFP protein levels, indicating glucose restriction promoted Hmg1 protein accumulation (*Figure 4A*). Interestingly, *nvj1Δ* yeast displayed a more marked increase in Hmg1-GFP protein compared to wildtype. To determine the influence of proteasomal degradation on Hmg1 protein levels during AGR, we treated cells with proteasome inhibitor MG132 and observed only a moderate increase in steady state Hmg1-GFP protein levels (*Figure 4B*). In contrast, treatment with the translation inhibitor cycloheximide during AGR returned Hmg1-GFP protein abundance to levels comparable with log-phase cells. Collectively, this analysis suggests that proteasomal degradation does not contribute significantly to controlling Hmg1-GFP protein abundance during AGR, and the increase is likely due to de novo protein synthesis during glucose restriction.

To further explore the metabolic cues governing AGR-induced Hmg1 synthesis and spatial partitioning, we monitored Hmg1-GFP localization in yeast lacking the major glucose-sensing kinase Snf1, the yeast AMPK homolog that regulates metabolic remodeling during changes in glucose availability (*Hedbacker and Carlson, 2008*). Surprisingly, *snf1Δ* yeast maintained AGR-induced Hmg1-GFP partitioning at the NVJ, indicating Hmg1-GFP recruitment to the NVJ was not dependent on Snf1/AMPK signaling (*Figure 4C*). We next examined whether Hmg1-GFP NVJ partitioning required the ergosterol-sensing transcription factor Upc2 that controls yeast sterol synthesis in a manner similar to mammalian SREBP signaling (*Yang et al., 2015*). Indeed *upc2Δ* yeast failed to partition Hmg1-GFP at the NVJ during AGR, and this was specific to *upc2Δ*, as Hmg1-GFP maintained NVJ partitioning in *ecm22Δ* yeast, a Upc2 paralog (*Figure 4D–F*). To determine whether loss of Upc2 was influencing Hmg1 partitioning through Nvj1, we imaged Nvj1-mNg in wild type or *upc2Δ* cells. We found that neither Nvj1 localization, Nvj1 expression, nor NVJ size was affected in the absence of Upc2 (*Figure 4—figure supplement 1A–C*); therefore, loss of Hmg1 NVJ partitioning in *upc2Δ* cells occurs independently of Nvj1. Additionally, we monitored Hmg1-GFP expression in *upc2Δ* cells and found no significant changes when compared to Hmg1-GFP wild type cells (*Figure 4—figure supplement 1D*). These data suggest that there are yet unidentified factors that are needed for Hmg1 NVJ partitioning that require the presence of Upc2, as well as the luminal region of Nvj1, and that partitioning might be uncoupled from Hmg1 protein expression.

Control of Hmg1 partitioning via Upc2 implied that Hmg1-GFP NVJ partitioning correlated with alterations in cellular ergosterol levels. In line with this, Hmg1-GFP NVJ partitioning was also induced by treatment with the squalene epoxidase inhibitor terbinafine, which blocks de novo ergosterol biogenesis (*Figure 4G–I*). To directly examine cellular sterols, we conducted thin layer chromatography (TLC). Indeed, following 4 hr of AGR when Hmg1-GFP is NVJ partitioned, yeast exhibited ~30% more steady-state levels of sterol-esters (SE), while free ergosterol levels were unchanged (*Figure 5A,B*). Co-treatment with the HMGCR inhibitor lovastatin during AGR suppressed this SE elevation, indicating the elevated SE pool originated from de novo SE synthesis rather than esterification of pre-existing ergosterol. Altogether, this suggests that AGR induces an Upc2-dependent compartmentalization of Hmg1 at the NVJ, which correlates with elevated de novo SE synthesis.

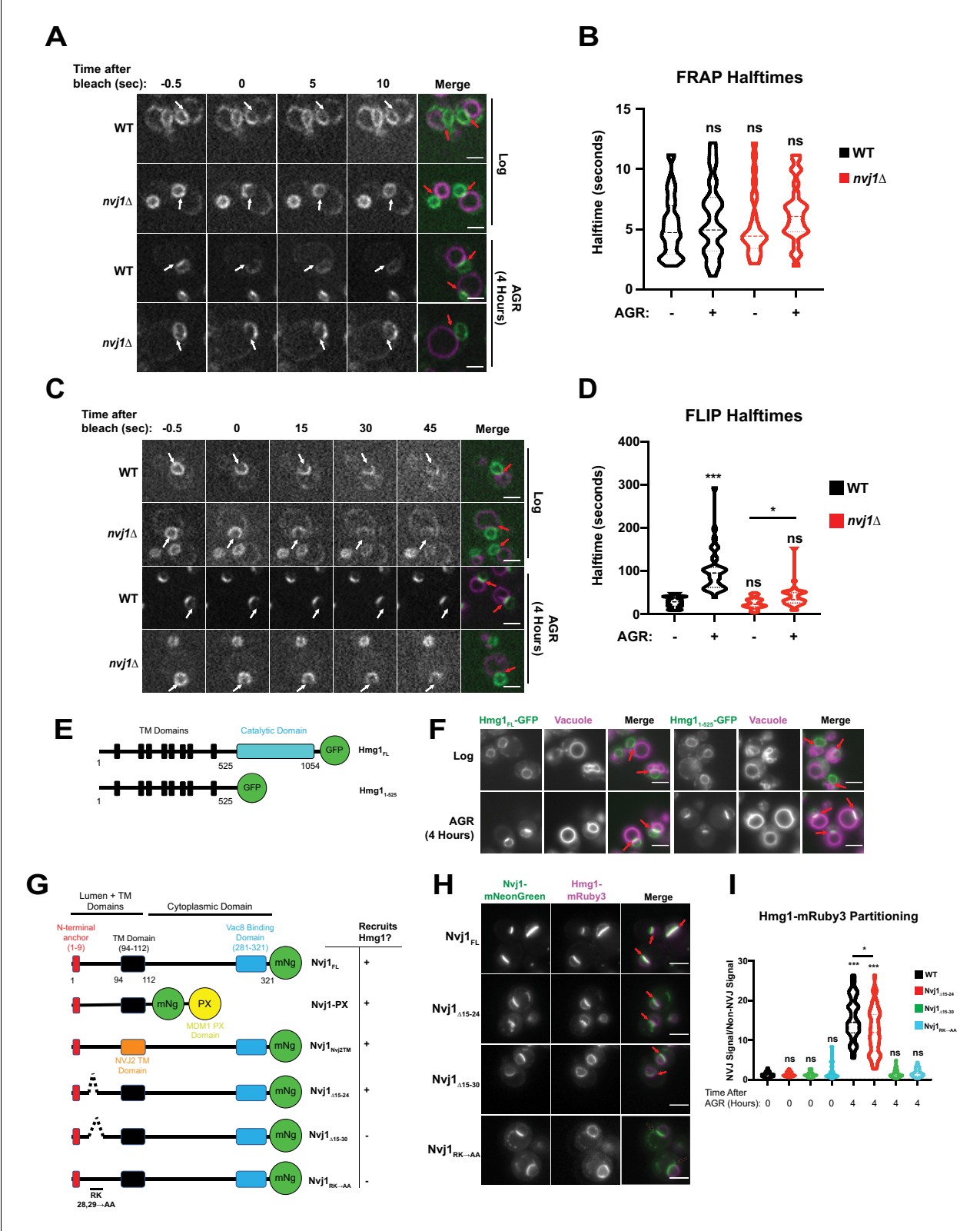

**Figure 3.** Hmg1 is selectively retained at the NVJ and requires a luminal domain of Nvj1. (**A**) Spinning disk confocal microscopy images from FRAP movies of yeast expressing endogenously tagged Hmg1-GFP and stained with FM4-64 (magenta). Portions of the NE corresponding to the NVJ were photobleached (white arrows) and recovery was monitored in both WT and *nvj1Δ* cells grown to log phase or after 4 hr of AGR. Red arrows represent the location of the NVJ. (**B**) Quantification of Hmg1-GFP FRAP halftimes after photobleaching. (Brown-Forsyth and Welch ANOVA. N > 26 cells) (**C**)

*Figure 3 continued on next page*

*Figure 3 continued*

Spinning disk confocal microscopy images taken from FLIP movies. Conditions and strains were the same as in (A). One region of the NE that lies opposite the NVJ was photobleached every 5 s (white arrows) and Hmg1-GFP signal at the NVJ was monitored for loss of fluorescence. Time above the images indicates seconds after the first bleach pulse. Red arrows represent the location of the NVJ (D) Quantification of FLIP halftimes. (Brown-Forsyth and Welch ANOVA. N > 29 cells. *p-value>0.05 ***p-value<0.001). (E) Cartoon of Hmg1-GFP constructs endogenously expressed in yeast with both the cytosolic catalytic domain (blue) and TM/luminal domains (black) (Hmg1$_{FL}$) or the TM/luminal domains alone (Hmg1$_{1-525}$). (F) Epifluorescence microscopy images of yeast expressing endogenously tagged Hmg1$_{FL}$-GFP or Hmg1$_{1-525}$-GFP. (G) Cartoon of Nvj1-mNeonGreen constructs expressed in *nvj1Δ* cells. Nvj1$_{FL}$-mNg contains all domains of Nvj1 including the N-terminal luminal anchor (red), the TM domain (black), and the Vac8-binding domain (VBD) (blue). Chimeric constructs either replaced the VBD with a vacuole-binding PX domain (Nvj1$_{PX}$) or replaced the endogenous TM domain with the TM domain of Nvj2 (Nvj1$_{NVJ2TM}$). Images of these constructs can be found in *Figure 3—figure supplement 1A*. Truncations of Nvj1 removed either residues 15–24 (Nvj1$_{15-24Δ}$), 15–30 (Nvj1$_{15-30Δ}$), or mutated residues 28 and 29 to alanine (Nvj1$_{RK→AA}$). (H) Epifluorescence microscopy images of yeast expressing truncation constructs depicted in (G). Cells were co-expressing endogenously tagged Hmg1-mRuby3 (magenta). Scale bars represent 5 μm. (I) Quantification of Hmg1 partitioning at the NVJ for yeast expressing truncations of Nvj1-mNg and Hmg1-mRuby3. (Brown-Forsyth and Welch ANOVA. N > 61 cells. *p-value<0.05 ***p-value<0.001).

The online version of this article includes the following figure supplement(s) for figure 3:

**Figure supplement 1.** Hmg1 NVJ partitioning is not dependent on the cytoplasmic or transmembrane regions of Nvj1.

## NVJ partitioning of HMGCR enhances mevalonate pathway flux

Given that *nvj1Δ* yeast accumulate more Hmg1-GFP protein than wild-type cells (*Figure 4A*), we were surprised to find that *nvj1Δ* yeast produce similar steady-state levels of SEs following AGR. This implied that HMGCR enzymes in *nvj1Δ* yeast could be catalytically less efficient. If true, we would expect: (1) accumulation of the HMGCR substrate HMG-CoA and (2) a decrease in downstream mevalonate pathway products such as squalene, ergosterol, and SEs. To interrogate whether Hmg1 spatial compartmentalization influenced Hmg1 enzymatic activity and/or mevalonate pathway flux, we used $^{14}$C-acetate pulse-radiolabeling to monitor these mevalonate pathway components. Indeed, *nvj1Δ* yeast exhibited significantly elevated $^{14}$C-labeled HMG-CoA after a 15-min radio-pulse, and contained significantly less $^{14}$C-labeled squalene, ergosterol, and SE (*Figure 5C–F*, *Figure 5—figure supplement 1A*). However, $^{14}$C-diacylglycerol (DAG) was not significantly impacted, suggesting these alterations were specific to mevalonate pathway flux and not due to a non-specific dilution of the $^{14}$C-acetate radiolabel in *nvj1Δ* cells (*Figure 5G*).

To investigate whether these changes in mevalonate metabolism affected yeast fitness or growth, we monitored yeast growth after a 10 hr period of AGR stress, followed by re-introduction to SC-media containing 2% glucose (SC+glucose). Indeed, *nvj1Δ* yeast displayed slower growth in culture following an AGR to SC+glucose transition compared to WT (*Figure 5H*). Notably, this was rescued by addition of exogenous mevalonate, suggesting the growth defect was attributed to defects in the mevalonate pathway (*Figure 5H*). To dissect the nature of this delayed cell growth, we conducted single-cell time-lapse imaging of yeast exposed to 10 hr of SC media lacking glucose (which induces cell cycle arrest and a halt to cell budding), followed by a glucose replenishment phase when yeast were again re-supplied with SC+glucose media, which promoted budding resumption. Quantifying the time required for growth resumption following this SC to SC+glucose transition, as measured by the appearance of the first daughter bud, showed that *nvj1Δ* cells had a significant delay in growth resumption (*Figure 5I*). In line with this, doubling times of cells were not affected by either loss of Nvj1 or addition of mevalonate following growth resumption (*Figure 5—figure supplement 1B*). Together, these findings indicate that *nvj1Δ* yeast unable to spatially compartmentalize Hmg1 manifest alterations in mevalonate pathway flux, and that *nvj1Δ* yeast manifest growth defects following AGR that can be rescued by exogenous mevalonate addition.

## Altered mevalonate pathway flux is not due to the general loss of the NVJ

To rule out the possibility that complete ablation of the NVJ contact site itself was responsible for reduced mevalonate pathway flux in *nvj1Δ* cells, we subjected *nvj1Δhmg2Δ* cells expressing either Nvj1$_{FL}$ or Nvj1$_{RK→AA}$ to radio-pulse analysis of the mevalonate pathway. Because the Nvj1$_{RK→AA}$ mutant still formed NVJs but did not recruit Hmg1, we were able to more precisely determine the role of Hmg1 partitioning specifically on mevalonate pathway flux. As seen for *nvj1Δ* cells, Nvj1$_{RK→AA}$ cells accumulated labeled HMG-CoA after a short pulse with $^{14}$C-acetate (*Figure 6A*). Furthermore,

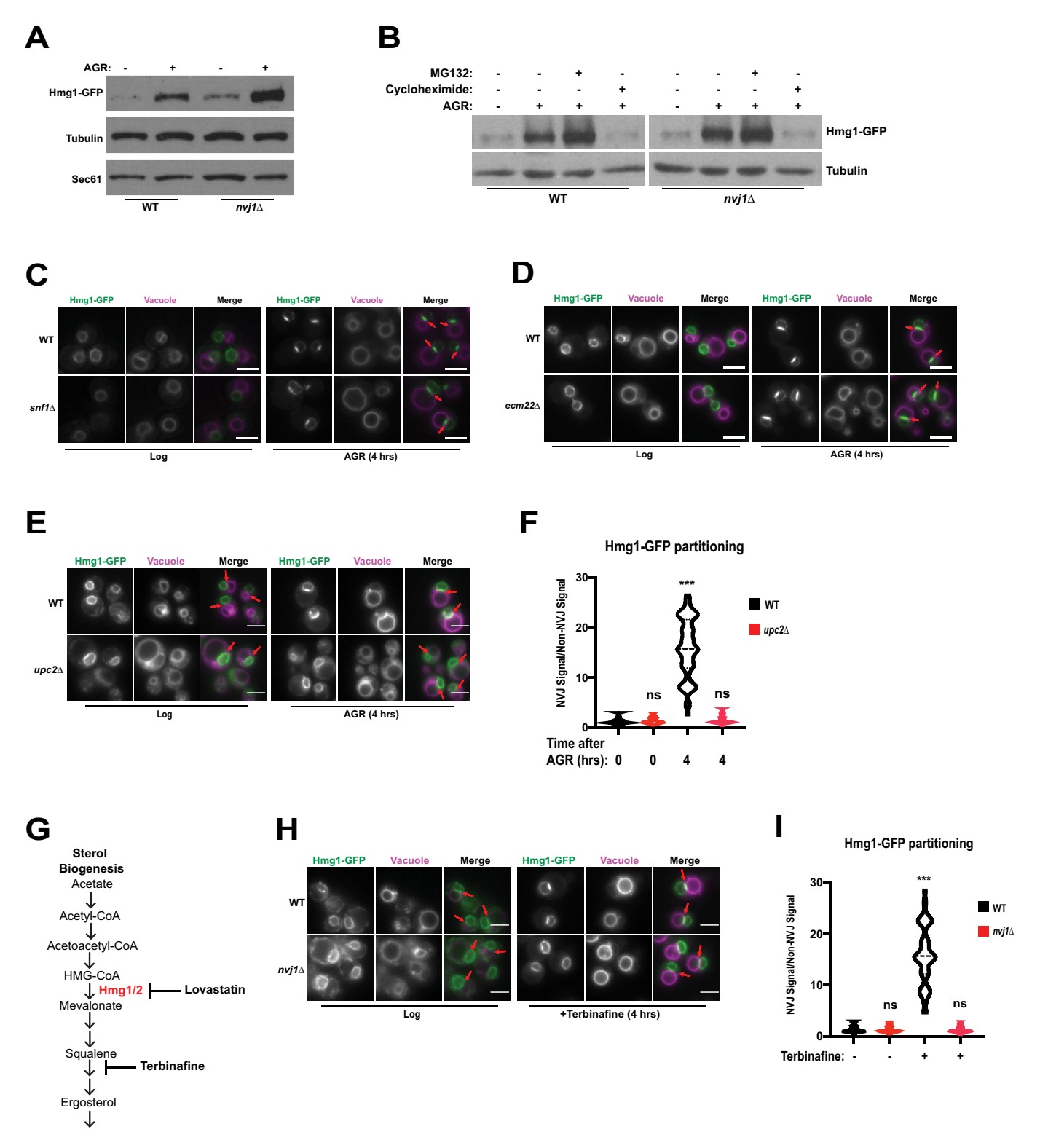

**Figure 4.** Hmg1 protein accumulates in *nvj1Δ* yeast via enhanced synthesis, which coincides with Upc2-dependent Hmg1 clustering at the NVJ. (**A**) Immunoblot of cells expressing endogenously tagged Hmg1-GFP grown to log-phase or treated with AGR for 4 hr. Tubulin and Sec61 antibodies were used as loading controls. (**B**) Immunoblot of Hmg1-GFP expressing cells grown to log-phase or treated with AGR for 4 hr. AGR-treated cells were also co-treated with either 100 μg/mL cycloheximide or 25 μM MG132. Tubulin was used as a loading control. (**C**) Epifluorescence microscopy images of cells expressing endogenously tagged Hmg1-GFP in either a wild type (WT) or Snf1 knock-out (*snf1Δ*) background. (**D**) Epifluorescence microscopy
*Figure 4 continued on next page*

*Figure 4 continued*

images of cells expressing endogenously tagged Hmg1-GFP in either a wild type (WT) or *ecm22Δ* background. Scale bars = 5 µm. (E) Epifluorescence microscopy images of endogenously tagged Hmg1-GFP (green) with FM4-64-stained vacuoles (magenta) grown to log phase or after 4 hr of AGR in wild type (WT) and *upc2Δ* yeast. Scale bars = 5 µm. Red arrows indicate relative position of the NVJ. (F) Quantification of Hmg1-GFP partitioning at NVJ from images represented in (E). (Brown-Forsyth and Welch ANOVA. N > 49 cells. ***p-value<0.001). (G) Schematic representing abbreviated ergosterol biogenesis pathway. (H) Epifluorescence microscopy images of wild type and *nvj1Δ* cells expressing endogenously tagged Hmg1-GFP treated with 10 µg/mL of terbinafine for 4 hr. Scale bars represent 5 µm. (I) Quantification of Hmg1-GFP partitioning at the NVJ from images shown in (H). (Brown-Forsyth and Welch ANOVA. N > 51 cells. ***p-value<0.001).

The online version of this article includes the following figure supplement(s) for figure 4:

**Figure supplement 1.** Loss of Hmg1 partitioning in *upc2Δ* cells does not function through Nvj1 or Hmg1 expression.

[14]C-labeled squalene, ergosterol, and sterol-esters were significantly decreased in the Nvj1$_{RK\rightarrow AA}$ mutant, while DAG production was unaffected (*Figure 6B–D*; *Figure 6—figure supplement 1A,B*). We also observed by single-cell time lapse imaging that the Nvj1$_{RK\rightarrow AA}$ mutant cells experience growth resumption delay following an SC to SC+glucose media transition, similar to *nvj1Δ* cells (*Figure 6E,F*). It is unlikely that these effects can be contributed to different Hmg1 expression in the given backgrounds, as Hmg1-mRuby3 fluorescence intensity in Nvj1$_{FL}$ and Nvj1$_{RK\rightarrow AA}$ cells were comparable during AGR (*Figure 6—figure supplement 1C*). Overall, this further supports a model where Hmg1 partitioning at the NVJ, and not simply the presence of an NVJ contact, contributes to increased mevalonate pathway flux and resumption of growth following glucose starvation.

## Uncoupling Hmg1 abundance from compartmentalization indicates a role for NVJ partitioning in mevalonate pathway flux

The AGR-induced increases in Hmg1 protein abundance made it more challenging to specifically dissect the role of Hmg1 spatial compartmentalization in mevalonate metabolism. To mechanistically dissect this compartmentalization and uncouple it from Hmg1 protein level, we generated yeast strains lacking endogenous Hmg1 and Hmg2 and expressing Hmg1-GFP from a non-native *ADH* promoter (*Figure 6G*). This new strain (ADHpr:Hmg1-GFP) maintained similar Hmg1-GFP protein levels with and without AGR treatment, and in the presence or absence of Nvj1 (*Figure 6H*). Critically, ADHpr:Hmg1-GFP yeast still partitioned Hmg1-GFP at the NVJ during AGR in an Nvj1-dependent manner, indicating we had uncoupled Hmg1 protein levels from Hmg1-GFP spatial compartmentalization (*Figure 6I*).

Next, we dissected whether loss of NVJ spatial compartmentalization in this ADHpr:Hmg1-GFP yeast impacted mevalonate pathway flux. Radio-pulse analysis revealed that *nvj1Δ* yeast with ADHpr:Hmg1-GFP displayed exacerbated alterations in mevalonate pathway flux, exhibiting elevated [14]C-labeled HMG-CoA levels and decreased [14]C-squalene, ergosterol, and SEs (*Figure 6J–M*, *Figure 6—figure supplement 1D*). Once again, [14]C-DAG levels were unaffected, indicating that *nvj1Δ* yeast manifested specific defects in mevalonate metabolism and not other ER-associated lipid metabolic pathways (*Figure 6—figure supplement 1E*). This suggests that Hmg1-GFP NVJ spatial partitioning, independent of Hmg1 protein levels, promotes mevalonate pathway flux.

## Loss of Hmg1 NVJ partitioning can be bypassed by compartmentalizing Hmg1 via artificial multimerization

Bioengineering studies indicate that HMGCR enzymatic domains exhibit enhanced catalytic activity when forced into close proximities via multi-valent flexible scaffolds (*Dueber et al., 2009*). Similarly, human HMGCR requires tetramerization for catalytic activity (*Istvan et al., 2000*). We hypothesized that Hmg1 NVJ partitioning may enhance enzymatic activity by promoting the probability of close physical associations between HMGCR enzymes as they are retained in the NVJ. Indeed, when we analyzed Hmg1-3HA using blue native PAGE (BN-PAGE) we found that Hmg1 associated in high-molecular-weight (HMW) species corresponding to an approximate molecular mass of 720 kDa (*Figure 7A,B*). Appearance of Hmg1 HMW species was dramatically increased during AGR, and HMW species were significantly reduced in *nvj1Δ* cells. Differences in HMW species abundance was not simply due to Hmg1-3HA expression, as the trend was observed even when Hmg1-3HA loading was adjusted evenly as observed by SDS-PAGE from the same samples (*Figure 7A,B*).

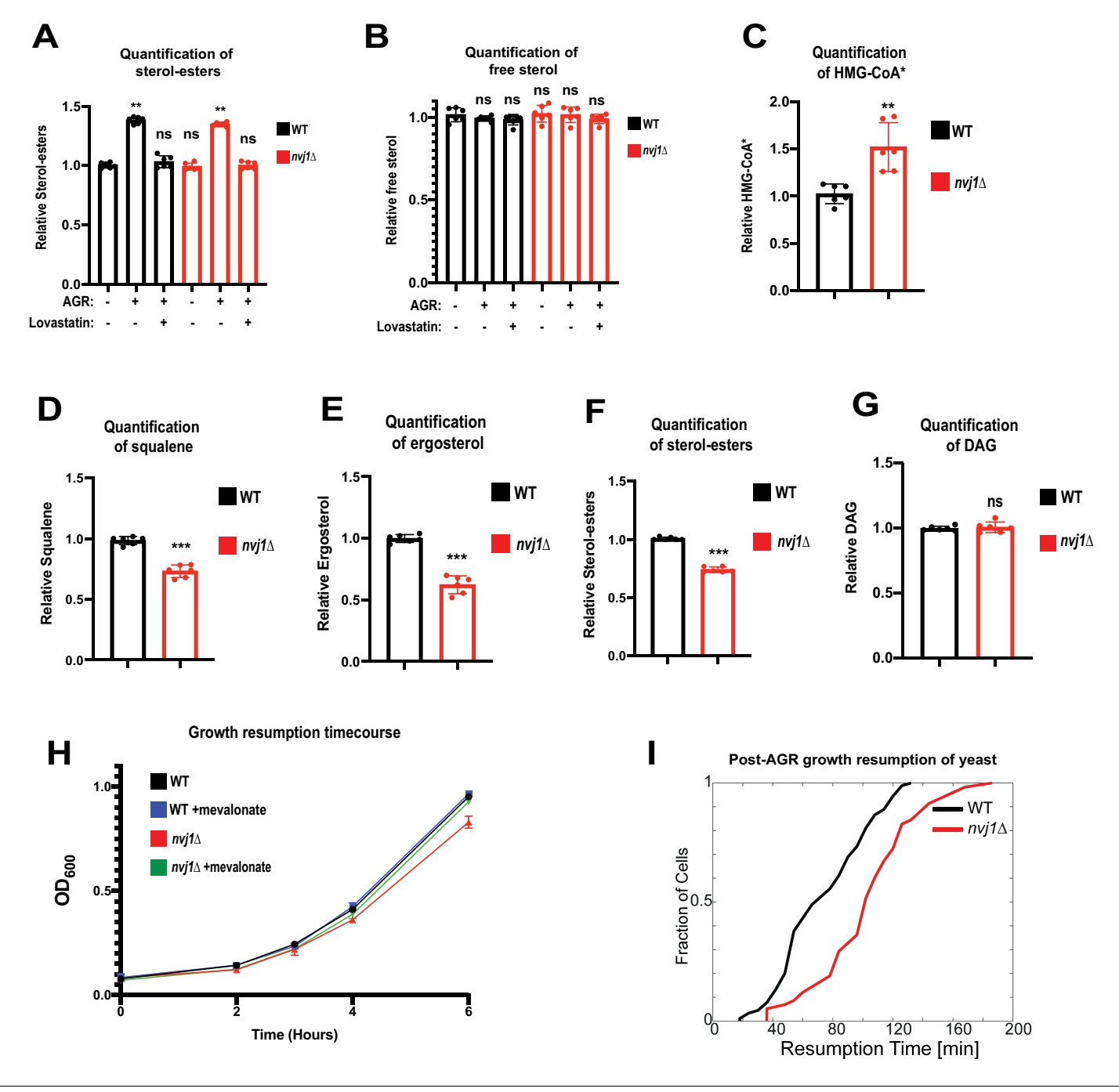

**Figure 5.** Hmg1 NVJ partitioning coincides with de novo sterol-ester production and increased Hmg1 catalytic efficiency. (**A**) Quantification of SEs in wild type and *nvj1Δ* cells collected at log phase, after 4 hr of AGR, or after a 4 hr co-treatment of AGR in the presence of 20 μg/mL lovastatin. Neutral lipids were extracted, separated by TLC, and visualized using Cu(II) Sulfate spray and charring. Quantification performed by densitometry. (Brown-Forsyth and Welch ANOVA. N = 6. **p-value<0.01) (**B**) Quantification of free sterols using the same strains and methodology as in (**A**). (Brown-Forsyth and Welch ANOVA. N = 6. ***p-value<0.001). (**C**) Scintillation counting quantification of radiolabeled HMG-CoA* produced after a 15 min pulse with [14]C-acetate in intact WT or *nvj1Δ* cells grown for 4 hr in AGR. After quenching the radio-labeling reaction, soluble metabolites were isolated and endogenous HMG-CoA was converted into mevalonate using an in vitro HMGCR reaction. Mevalonate was subsequently separated and isolated by TLC, visualized by autoradiography, and quantified by scintillation counting. (Brown-Forsyth and Welch ANOVA. N = 6. **p-value<0.01). (**D–G**) Quantifications of radiolabeled DAG and mevalonate-derived lipids. Radio-labeling was performed as described in (**C**). Lipids were extracted and separated by TLC and visualized by autoradiography. Squalene and SE bands were quantified by scintillation counting. Ergosterol and DAG bands were quantified by densitometry. (Brown-Forsyth and Welch ANOVA. N = 6. ***p-value<0.001). (**H**) Growth curves of WT and *nvj1Δ* cells. Cells were

*Figure 5 continued on next page*

*Figure 5 continued*

treated with AGR for 10 hr in the absence or presence of 10 µg/mL mevalonate, and subsequently diluted to $OD_{600}$ = 0.1 in SC media containing 2% glucose lacking mevalonate. $OD_{600}$ measurements were taken every hour following dilution in fresh glucose-containing media. (I) Resumption probability graph depicting probability of cells producing a daughter bud as a function of time following an SC to SC+glucose transition (N > 85 cells). The online version of this article includes the following figure supplement(s) for figure 5:

**Figure supplement 1.** Cells lacking endogenous Nvj1 accumulate Hmg1 substrates and have decreased mevalonate-derived pathway products.

To test whether loss of Hmg1 HMW species in *nvj1Δ* cells was contributing to lower mevalonate pathway flux, we fused Hmg1 to a constitutively tetrameric fluorescent protein, DsRed2, and determined whether this artificial multimerization could bypass the loss of NVJ-mediated compartmentalization by comparing it to Hmg1 tagged with a monomeric fluorescent protein, mRuby3 (*Figure 7C*). Both Hmg1-mRuby3 and Hmg1-DsRed2-tagged strains manifested similar increases in Hmg1 abundance in AGR, suggesting this tagging did not affect AGR-induced Hmg1 protein increase (*Figure 7—figure supplement 1D*). Furthermore, Hmg1-DsRed2 protein migrated at higher than expected molecular weight even in SDS-PAGE gels compared to Hmg1-mRuby3, suggesting the DsRed2 tag promoted self-association (*Figure 7—figure supplement 1E*). Hmg1-DsRed2 foci were also noted on the NE during AGR, consistent with stabilized Hmg1 multimers (*Figure 7—figure supplement 1A*).

As expected, *nvj1Δ* cells tagged with mRuby3 still manifested elevated $^{14}$C-labeled HMG-CoA and reduced squalene, ergosterol, and SE labeling associated with NVJ loss (*Figure 7D–G*). Strikingly, these perturbations were rescued in the *nvj1Δ* Hmg1-DsRed2 strain, closely mirroring wildtype $^{14}$C-labeled levels of Hmg-CoA, squalene, ergosterol, and SE (*Figure 7D–G*, *Figure 7—figure supplement 1B,C*). Once again, appending either mRuby3 or DsRed2 tags to Hmg1 did not significantly alter expression of the protein from what was observed with other tags, and our data cannot be explained by differences in Hmg1 expression alone (*Figure 7—figure supplement 1D*). Collectively, this indicates that artificial multimerization of Hmg1 can bypass the loss of NVJ-mediated Hmg1 compartmentalization.

Next, we interrogated whether Hmg1-DsRed2 tagging would rescue the growth resumption delay observed for *nvj1Δ* yeast following 10 hr exposure to SC media lacking glucose. As expected, growth resumption delay was observed in *nvj1Δ* yeast expressing Hmg1-mRuby3. Remarkably, expression of Hmg1-DsRed2 in *nvj1Δ* cells rescued this defect (*Figure 7H,I*). Collectively, this is consistent with a model where NVJ-dependent Hmg1 spatial compartmentalization via NVJ partitioning or artificial multimerization enhances mevalonate pathway flux as well as defects in growth resumption following glucose starvation.

## Discussion

Acute changes in nutrient availability trigger metabolic remodeling that is characterized by alterations in metabolic pathway flux and metabolite supply and demand. How cells spatially and temporally coordinate this remodeling remains poorly characterized, yet critical to our understanding to cell adaptation and survival. Changes in nutrient availability can drive the subcellular re-distribution of enzymes within cells, as well as the formation of enzyme assemblies or complexes that promote or fine-tune metabolic pathways. Here, we present evidence that AGR in yeast enhances mevalonate pathway flux, and propose the mechanism underlying this requires the spatial compartmentalization of the rate-limiting HMGCR enzymes at the yeast NVJ (*Figure 7J*).

Through time-lapse and FRAP/FLIP-based imaging, we find that Hmg1 is recruited and selectively retained at the NVJ in a manner requiring basic residues within the Nvj1 luminal domain. We also demonstrate that Hmg1 partitioning at the NVJ can be uncoupled from Nvj1 binding of Vac8, which further supports a model where Hmg1 partitioning is being controlled predominantly from the ER. One model is that Hmg1 is selectively recruited as an enzyme 'client' to the Nvj1 'scaffold' during AGR, but we cannot rule out the possibility that other factors may be involved together with Nvj1 in Hmg1 selective retention at the NVJ. Indeed, Upc2, a sterol-sensing transcription factor required for Hmg1 NVJ partitioning, appears to influence Hmg1 independent of either Nvj1 expression or localization, which may suggest additional factors required for Hmg1 partitioning at the NVJ. Both the

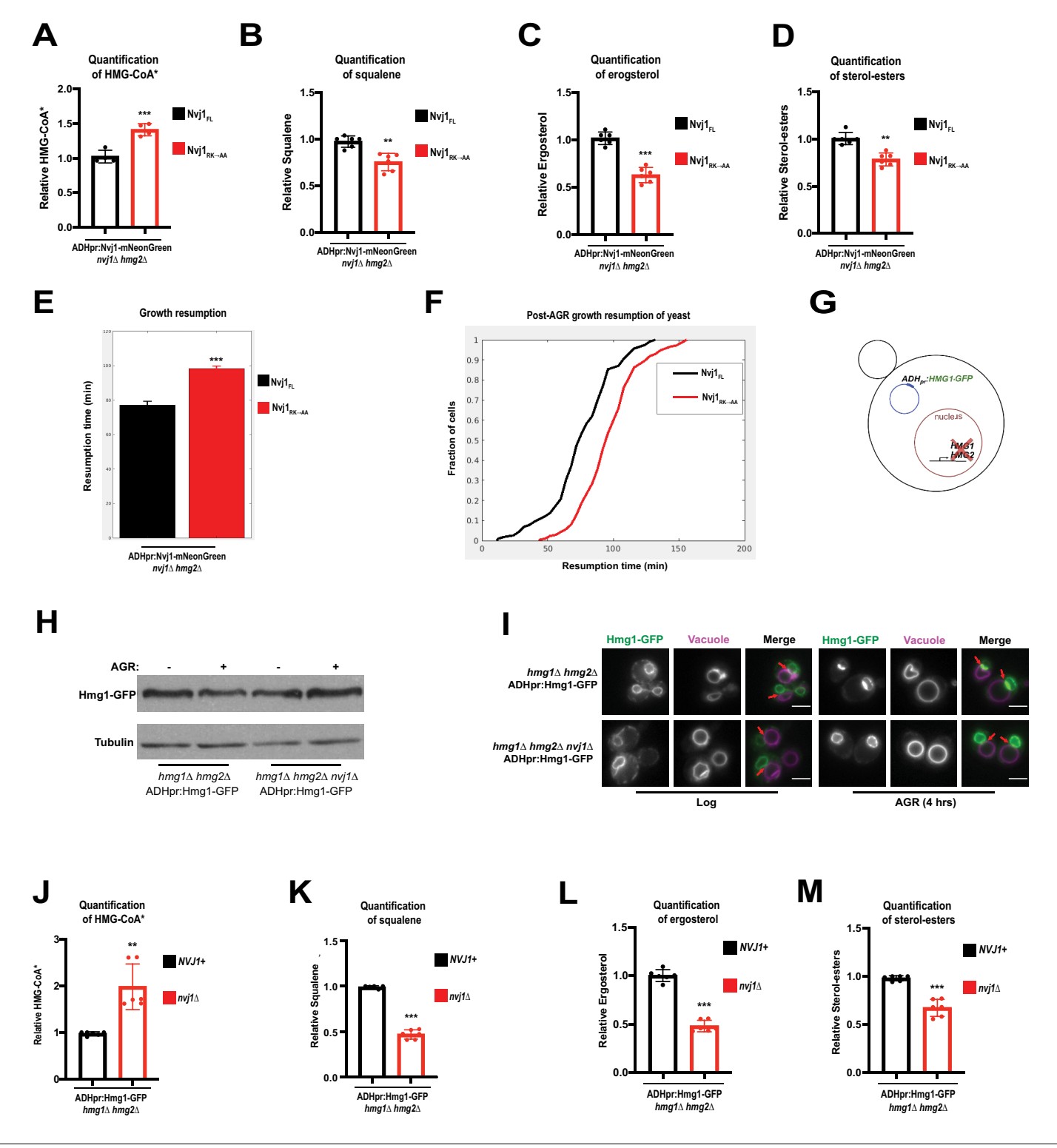

**Figure 6.** AGR-induced compartmentalization of Hmg1 increases enzymatic efficiency. (A) Quantification of radiolabeled HMG-CoA* from *nvj1Δhmg2Δ* yeast expressing either Nvj1_FL or Nvj1_RK→AA. Growth and radiolabeling was performed as described above. (Brown-Forsyth and Welch ANOVA. N = 6. ***p-value<0.001). (B–D) Quantification of radiolabeled squalene, ergosterol, and SE produced from a 15 min radio-labeling pulse in *nvj1Δhmg2Δ* yeast expressing either Nvj1_FL or Nvj1_RK→AA. (Brown-Forsyth and Welch ANOVA. N = 6. **p-value<0.01, ***p-value<0.001). (E) Quantification of growth resumption times during SC to SC+glucose transition in *nvj1Δhmg2Δ* yeast expressing either Nvj1_FL or Nvj1_RK→AA (N > 82 cells, ***p<0.0001, Kolmogorov-Smirnov test). (F) Resumption probability graph depicting probability of cells producing a daughter bud as a function of time following an

*Figure 6 continued on next page*

*Figure 6 continued*

SC to SC+glucose transition (N > 82 cells). (G) Cartoon representation of ADHpr:Hmg1-GFP strains. Strains were generated by removal of endogenous Hmg1 and Hmg2, and replacing with ADH promoter-driven Hmg1-GFP (ADHpr:Hmg1-GFP). (H) Immunoblot of Hmg1-GFP isolated from ADHpr:Hmg1-GFP strains before or after treatment with AGR. Tubulin was used as a loading control. (I) Epifluorescence imaging of ADHpr:Hmg1-GFP strains showing that Hmg1 NVJ partitioning can be uncoupled from endogenous protein expression. Red arrows indicate relative position of the NVJ. Scale bars = 0.5 µm. (J) Quantification of radiolabeled HMG-CoA* from yeast expressing ADHpr:Hmg1-GFP in wildtype (*NVJ1+*) or *nvj1Δ* yeast. Growth and radiolabeling was performed as described above. (Brown-Forsyth and Welch ANOVA. N = 6. **p-value<0.01). (K–M) Quantification of squalene, ergosterol, and SE produced from a 15 min radio-labeling pulse in ADHpr:Hmg1-GFP strains. (Brown-Forsyth and Welch ANOVA. N = 6. ***p-value<0.001).

The online version of this article includes the following figure supplement(s) for figure 6:

**Figure supplement 1.** Cells expressing ADHpr:Hmg1-GFP in the absence of endogenous Nvj1 have exacerbated Hmg1 catalytic deficiency phenotype.

requirement of Upc2 and the sufficiency of the Erg1 inhibitor terbinafine to induce Hmg1 NVJ partitioning is consistent with AGR stress producing a cellular demand for mevalonate. In line with this, we observe an increase in de novo SE synthesis during AGR; however, we cannot rule out that other mevalonate-derived metabolites may play important roles during AGR. In fact, we would expect AGR to promote a metabolic switch to mitochondrial respiration, which in part utilizes at least one mevalonate-derived metabolite, coenzyme Q, to function as an electron carrier. In support of this, we do not observe significant decline of steady-state SE levels in *nvj1Δ* cells, suggesting that mevalonate may be shunted into several pathways during AGR. Remarkably, *nvj1Δ* yeast manifest alterations in mevalonate pathway flux and growth defects when faced with glucose starvation, which can be rescued by the addition of exogenous mevalonate (the HMGCR enzymatic product).

We have additionally uncovered that mevalonate pathway flux can be rescued in *nvj1Δ* cells by artificially multimerizing Hmg1 using a tetrameric fluorophore. In line with this, we have observed that Hmg1 has a propensity to assemble into HMW species > 720 kDa during AGR in a Nvj1-dependent manner. Thus, mevalonate pathway flux during AGR appears tightly correlated with the ability to form HMW Hmg1 species at the NVJ, rather than the total Hmg1 protein level in the cell. To more fully dissect the role of HMGCR spatial compartmentalization on mevalonate metabolism, we generated an artificial yeast strain that maintains more constant Hmg1 protein levels and inducibly accumulates at the NVJ, thus functionally uncoupling Hmg1 protein abundance from NVJ spatial compartmentalization. Remarkably, this strain exhibits defects in mevalonate pathway flux when Hmg1 cannot be NVJ partitioned, again underscoring the role of Hmg1 spatial compartmentalization in fine-tuning mevalonate metabolism. Collectively, we present a model where AGR-induced metabolic remodeling of mevalonate metabolism is spatially coordinated at the NVJ via selective retention of HMGCRs and their subsequent incorporation into HMW species. An intriguing possibility is that, in general, inter-organelle contact sites act as platforms that slow the diffusion of proteins that enter them via their interactions with resident tether 'scaffolds', thus increasing the local concentrations of proteins and enzymes sufficient to fine-tune metabolic flux.

Glucose and other nutrient limitations can induce autophagy and have previously been shown to induce micro-lipophagy, where lipid droplets are engulfed by the yeast vacuole (*Seo et al., 2017*). Indeed, we have observed that LDs accumulate in close proximity to the NVJ during periods of glucose limitation (*Hariri et al., 2018*; *Hariri et al., 2019*). The relationship between LD biogenesis and HMGCR partitioning at the NVJ is not fully resolved here and requires further study. An intriguing possibility is that the spatial re-organization of HMGCRs may contribute to LD biogenesis or compositional remodeling during the general metabolic remodeling required during glucose restriction.

In addition to inducing the compartmentalization of HMGCRs at the NVJ, yeast glucose starvation has also been shown to induce the formation of several cytoplasmic enzyme assemblies involved in nucleotide (Ura7), amino acid (Gly1), and lipid (Acs1) metabolism (*Munder et al., 2016*). These cytoplasmic assemblies are thought to reduce enzymatic activity and promote the transition into cellular dormancy. In contrast, other protein assemblies promote enzymatic activity. In the liver, the protein Mig12 binds to cytosolic acetyl-CoA carboxylases (Acc) and promotes their polymerization and catalytic activity, which elevates fatty acid synthesis and eventual triglyceride accumulation in hepatocytes (*Kim et al., 2010*). Our work provides evidence that HMGCRs form similar spatial assemblies at yeast ER-lysosome contact sites, implicating a role for inter-organelle junctions in the spatial

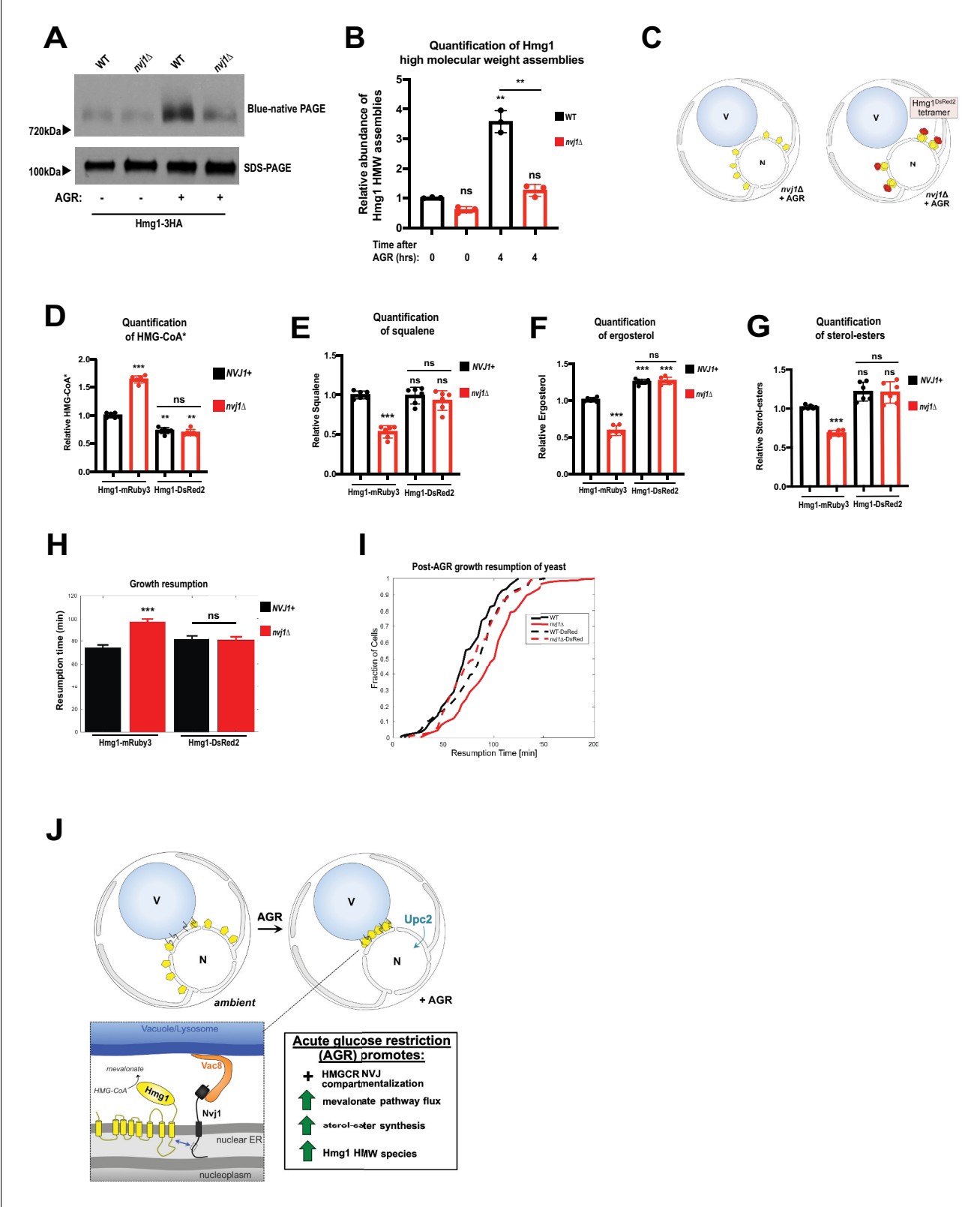

**Figure 7.** Artificial multimerization of Hmg1 rescues enzymatic deficiencies of *nvj1Δ* cells. (**A**) Blue native PAGE (BN-PAGE) and SDS-PAGE analysis of Hmg1-3HA expressed in either WT or *nvj1Δ* cells under log-phase or AGR growth conditions. Samples were loaded to mitigate differences in total Hmg1 loading such that differences in the BN-PAGE could be easily interpreted. Arrows on the side indicate relative molecular mass in corresponding region as determined by molecular weight standards. (**B**) Quantification of BN-PAGE and SDS-PAGE analysis as shown in (**A**). For quantification of

*Figure 7 continued*

relative Hmg1-3HA abundance in HMW species, fold change relative to WT log samples were first calculated for BN-PAGE species and subsequently normalized by total expression of Hmg1 in each sample/condition as determined by SDS-PAGE. (Brown-Forsyth and Welch ANOVA. N = 3. **p-value<0.01). (C) Cartoon representation of Hmg1 artificial tetramerization by DsRed2 tagging in *nvj1Δ* cells. (D) Scintillation counting quantification of radiolabeled HMG-CoA* from yeast expressing endogenously tagged monomeric Hmg1-mRuby3 or tetrameric Hmg1-DsRed2 in wildtype (*NVJ1+*) or *nvj1Δ* yeast. Quantification performed as in (*Figure 6A*). (Brown-Forsyth and Welch ANOVA. N = 6. **p-value<0.01 ***p-value<0.001). (E–G) Quantification of radiolabeled squalene, ergosterol, and SE produced in Hmg1-mRuby3 or Hmg1-DsRed2 yeast. Radio labeling and quantifications performed as in (*Figure 6B–D*). (Brown-Forsyth and Welch ANOVA. N = 6. ***p-value<0.001). (H) Average growth resumption times measured by single-cell time lapse microscopy. (WT: N = 110, nvj1Δ: N = 175, WT Hmg1-DsRed2: N = 94, nvj1Δ Hmg1-DsRed2: N = 96). (N > 94 cells, **p<0.001, Kolmogorov-Smirnov test). (I) Resumption probability graph depicting probability of cells producing a daughter bud as a function of time following an SC to SC+glucose transition. (N > 94 cells). (J) Cartoon representation of Nvj1 and Upc2-mediated Hmg1 partitioning at the NVJ. Spatial partitioning of Hmg1 at the NVJ coincides with increased production of mevalonate-derived metabolites.

The online version of this article includes the following figure supplement(s) for figure 7:

**Figure supplement 1.** Artificial inter-enzyme association via DsRed2 tagging rescues *nvj1Δ* associated Hmg1 catalytic deficiency.

coordination of metabolically-induced enzymatic assembly formation. With this, we have added to a long, and growing, list for roles fulfilled by inter-organelle contact sites in metabolic regulation.

## Materials and methods

### Key resources table

| Reagent type (species) or resource | Designation | Source or reference | Identifiers | Additional information |
|---|---|---|---|---|
| Strain, strain background (*Saccharomyces cervisiae*) | W303 | This paper | W303 | See *Supplementary file 1* for all yeast strains |
| Transfected construct (*Saccharomyces cervisiae*) | pRS305 vector backbone (Leu selection) | This paper | pRS305 | See *Supplementary file 2* for all yeast expression plasmids |
| Antibody | Anti-GFP (rabbit polyclonal) | Abcam | Cat# abcam290 | WB (1:10,000) |
| Antibody | anti-mRuby3 (rabbit polyclonal) | Invitrogen | Cat# R10367 | WB (1:1,000) |
| Antibody | anti-DsRed2 (mouse monoclonal) | Origene | Cat#: TA180005 | WB: (1:1,000) |
| Antibody | Anti-tubulin (rat monoclonal) | Abcam | Cat#: ab6160 | WB: (1:15,000) |
| Antibody | Sec61 (rabbit poly) | J. Friedman lab (UTSW) | N/A | WB: (1:15,000) |
| Software, algorithm | Segmentation and tracking code for single-cell time-lapse imaging of yeast budding | from N. E. Wood, A. Doncic, A fully-automated, robust, and versatile algorithm for long-term budding yeast segmentation and tracking. PLoS ONE. 14, e0206395 (2019). *Folch et al., 1957* | — | |

### Strains, plasmids, and yeast growth conditions

W303 (leu2-3,112 trp1-1 can1-100 ura3-1 ade2-1 his3-11,15) was used as the wild-type parental strain for all experiments and cloning in this study. All strains and plasmids can be found in *Supplementary files 1* and *2*, respectively. Deletion of endogenous genes, C-terminal tagging at endogenous loci, and transformation of plasmids was accomplished by the traditional lithium acetate method. Endogenous knockins and knockouts were validated by genomic PCR, and plasmids were

validated by sequencing. Plasmids generated for this study were created using RepliQa HiFi assembly (Quantabio cat. 95190) following the manufacturer's protocol. For cloning into pRS305 plasmids, destination vectors were cut with XhoI and SacI enzymes, while linearization of pFa6a plasmids was accomplished by digestion with AscI and PacI enzymes. Synthetic complete media supplemented with amino acids was used for all experiments, except where leucine was omitted from the media to accommodate growth of strains carrying pRS305 plasmids. All yeast were grown in 30°C incubators shaking at 210 RPM. Log phase yeast were grown to an $OD_{600}$ of 0.5 in the presence of 2% dextrose. AGR-treated yeast were grown to $OD_{600}$ of 0.5 in the presence of 2% dextrose, collected, washed with dextrose-free media, and resuspended in media containing 0.001% dextrose for the indicated time period. For experiments where TG lipolysis was induced, cells were grown to stationary phase over 24 hr and then diluted to an $OD_{600}$ of 0.5 in the presence of 2% dextrose and 10 μg/mL Cerulenin (Sigma C2389). Cells were incubated with cerulenin for 4 hr prior to collection. Where indicated, Terbinafine (Sigma T8826), Lovastatin (Sigma 1370600), Cycloheximide (Sigma C7698), or MG132 (Sigma M7449) were added to the media at the beginning of AGR treatment to final concentrations of 10 μg/mL, 20 μg/mL, 100 μg/mL, and 25 μM, respectively.

## Fluorescence microscopy

For confocal microscopy, cells were grown as described above, collected by centrifugation at 3000xg for 2 min, and resuspended in glucose-free media at approximately one one-hundredth of original volume. All images were taken as single slices at approximately mid-plane using a Zeiss LSM880 inverted laser scanning confocal microscope equipped with Zen software. Images were taken with a 63x oil objective NA = 1.4 at room temperature. Prior to imaging, cells were incubated for three hours with 0.5 μg/mL FM4-64 dye (Invitrogen T13320) to visualize vacuoles.

For epifluorescence microscopy, cells were grown, stained, and collected as described above. Vacuoles were also stained with 5 μg/mL CMAC (ThermoFisher C2110) dye for two hours prior to imaging, where indicated. Imaging was performed on an EVOS FL Cell Imaging System at room temperature. Hmg1 NVJ partitioning was quantified using Fiji software. For quantification, RGB images were converted to 16-bit and a background subtraction was performed by subtracting original images by a duplicate 'Gaussian blur' filtered image (sigma (radius) = 5.0). Five-pixel line scans were taken across the nuclear envelope toward the NVJ, and the 'plot profile' function was used in Fiji to produce a fluorescence histogram of nuclear envelope signal. The sum area under each curve was calculated and plotted as the ratio of fluorescence intensity of NVJ-associated signal by fluorescence intensity of Non-NVJ associated NE signal.

## FRAP and FLIP analysis

Yeast used for FRAP and FLIP were grown and collected as described above, and imaging was conducted for 1 hr after collection. Photobleaching movies were taken on an Andor spinning disk confocal microscope through a 63x oil objective (NA = 1.4). The microscope is equipped with an Andor Ultra EMCCD and Metamorph software. For FRAP measurements, a single circular ROI of 0.77 μm$^2$ that corresponds to the NVJ was selected and bleached with a 408 nm laser at 100% power and 100 ms dwell time. One image was taken before the bleach, and subsequent images were taken every 500 ms for a total movie length of 25 s. For FLIP measurements, single circular ROIs of 0.77 μm$^2$ were selected, taking care to select an area of the NE that was furthest from the NVJ. Each bleaching cycle consisted of a pre-bleach image, a single bleach with a 408 nm laser with a 100 ms dwell time, and four post-bleach images taken 500 ms apart. Each movie captured a total of 50 bleach cycles, which corresponds to 300 s movies. Fiji software was used to quantify bleaching curves and halftimes. Pre-processing of images included background subtraction as described above and 3D Gaussian smoothing (sigma = 0.5). FRAP quantification was performed using the double normalization method as previously described (**Phair and Misteli, 2000**). Briefly, intensity was measured for all time points in an ROI corresponding to the bleached region ($I_{frap}$) and an ROI corresponding to the whole cell ($I_{whole-cell}$), and normalized intensities were generated for each timepoint ($I_{normalized}(t)$) using **Equation 1**:

$$I_{normalized}(t) = \frac{I_{whole-cell-pre}}{I_{whole-cell}(t)} * \frac{I_{frap}(t)}{I_{frap-pre}} \qquad (1)$$

In the equation above, the 'pre' subtext indicates the timepoint preceding ROI bleaching. Intensity recovery curves were created for each movie by further normalizing values such that pre-bleach intensities were set to one and post-bleach intensity of an ROI was set to 0. Full normalization was accomplished using *Equation 2* followed by subtracting the normalized $I_{frap-bleach}$ value from all timepoints.

$$I_{normalized-full}(t) = \frac{I_{normalized}(t)}{I_{frap-pre} - I_{frap-bleach}}$$ (2)

In *Equation 2*, $I_{frap-bleach}$ indicates the intensity of the photobleached ROI at the time of photobleaching. Halftimes were calculated from individual fluorescence recovery curves using Graphpad Prism eight software and fitting the data to a one-phase exponential association. Pre-processing for FLIP images was the same as for FRAP images. FLIP movies were quantified using Fiji to monitor the intensity of NVJ-associated signal over time. Intensity measurements were normalized using *Equation 3*.

$$I_{FLIP}(t) = \frac{I_{NVJ}(t)}{I_{NVJ-pre-bleach}}$$ (3)

In *Equation 3*, $I_{FLIP}(t)$ is the relative fluorescence at the NVJ at time t, $I_{NVJ}(t)$ is the raw intensity value at the NVJ at time t, and $I_{NVJ-pre-bleach}$ is the intensity at the NVJ before bleaching occurred. Halftimes were calculated from FLIP decay curves, which were generated in Graphpad Prism eight software by fitting the data to a one-phase exponential decay.

## Lipid extraction and thin layer chromatography

For lipid extraction, approximately 50OD units of cells were collected for each sample, and pellet wet weight was normalized and noted prior to extraction. Lipid extraction was performed using a modified Folch method (*Folch et al., 1957*). Briefly, cell pellets were resuspended in MilliQ water with glass beads and lysed by three one-minute cycles on a bead beater. Chloroform and methanol were added to the lysate to achieve a 2:1:1 chloroform:methanol:water ratio. Samples were vortexed, centrifuged to separate the organic and aqueous phases, and the organic phase was collected. Extraction was repeated a total of three times. Prior to thin layer chromatography, lipid samples were dried under a stream of argon gas and resuspended in 1:1 chloroform:methanol to a final concentration corresponding to 4 μL of solvent per 1 mg cell pellet wet weight. Isolated lipids were spotted onto heated glass-backed silica gel 60 plates (Millipore Sigma 1057210001), and neutral lipids were separated in a mobile phase of 80:20:1 hexane:diethyl ether:glacial acetic acid. TLC bands were visualized by spraying dried plates with cupric acetate in 8% phosphoric acid and baking at 140°C for an hour. To quantify TLC bands, all plates were run with an internal neutral lipid standard. Densitometry of bands was performed in Fiji.

## Batch culture growth curves

Cells were treated with AGR for 10 hr, as described above. Some samples were also co-treated with 10 μg/mL mevalonate (Sigma 50838) at the beginning of AGR treatment. After a 10-hr treatment with AGR, cultures were diluted to an $OD_{600}$ of 0.1 in SC media containing 2% glucose. The $OD_{600}$ of cultures was measured each hour and plotted in Prism eight software.

## Single-cell time-lapse microscopy

Cells were imaged using a Zeiss Observer Z1 microscope equipped with automated hardware focus, motorized stage, temperature control, a Zeiss EC Plan-Apochromat 63 × 1.4 or 40 × 1.3 oil immersion objectives, and an AxioCam HRm Rev three camera. Exposure times for experiments shown in *Figure 5—figure supplement 1B*: Phase contrast 20 ms, Whi5-mKoκ 150 ms, Erg6-mTFP1 20 ms, Vma1-mNeptune2.5 75 ms, Nvj1-mRuby3 75 ms, Msn2-mNeonGreen 40 ms. Exposure times for experiments shown in *Figure 4* and *Figure 7—figure supplement 1*: Phase contrast: 40 ms, Hmg1-mRuby3 or Hmg1-DsRed2: 100 ms.

All experiments were performed with a Y04C Cellasic microfluidic device (http://www.cellasic.com/) using 1 psi flow rate. Images are taken every 6 min. Prior to loading into the microfluidics chamber, cells were sonicated and mixed with 50 μL SCD media to achieve an $OD_{600}$ of

approximately 0.1. In the chamber, cells are first grown for 2 hr in SCD. Next, they are exposed to acute glucose restriction (AGR) by switching to SC for 10 hr which is followed by glucose replenishment by 4 hr SCD. To determine time of growth resumption, cells are segmented and tracked as described previously (*Doncic et al., 2013*). Next, the time of resumption is annotated semi-automatically using a custom MATLAB software by determining the time of bud growth or new bud emergence during the 4-hr glucose replenishment following AGR.

## Immunoblotting

Approximately 50OD units of cells were collected for protein extraction. Prior to protein extraction, cell pellet wet weights were normalized. Protein extraction was accomplished by precipitating proteins with 20% TCA for 30 min on ice, followed by three washes of the pellet with cold 100% acetone. The protein pellet was dried for fifteen minutes in a speed-vac to remove residual acetone, and all pellets were resuspended in 2x SDS sample buffer (65.8 mM Tris-HCl, pH 6.8; 2% SDS; 25% glycerol; 10% 2-mercaptoethanol; 0.01% bromophenol blue). Resuspended protein samples were heated at 70°C for 10 min prior to being loaded onto a homemade 4–15% polyacrylamide gel and separated by electrophoresis. Proteins were transferred to a 0.45 μm nitrocellulose membrane in Towbin SDS transfer buffer using a Criterion tank blotter with plate electrodes (BioRad 1704070). Immediately following transfer, membranes were stained with PonceauS and cut using a clean razor blade. Membranes were blocked with 5% milk dissolved in TBS-T buffer, and primary antibodies were allowed to bind overnight at 4°C. Primary antibodies used for determining Hmg1 protein expression are as follows: GFP (Abcam ab290; 1:10,000 dilution), mRuby3 (Invitrogen R10367; 1:1000 dilution), and DsRed2 (OriGene TA180005; 1:1000 dilution). Rat monoclonal antibody to tubulin (Abcam ab6160; 1:15,000 dilution) and rabbit polyclonal antibody to Sec61 (Jonathan Friedman lab; 1:5000 dilution) were used as loading controls. Immunoblots were developed by binding HRP-conjugated anti-rabbit IgG (Sigma A0545; 1:10,000), anti-rat IgG (Abcam ab97057; 1:10,000), or anti-mouse IgG (Abcam ab6728; 1:2,000) secondary antibodies to the membrane for 1 hr in the presence of 5% milk followed by three washes in TBS-T and developing with ECL substrate (BioRad cat. 1705061). Signal was captured by X-ray film.

## Radiolabeling, metabolite extraction, and metabolite separation

Approximately 100OD units of cells were used for radiolabeling experiments. All cells were grown and treated with AGR as previously described. Prior to radiolabeling, cells were collected by centrifugation and washed with dextrose-free media. All liquid was removed from cell pellets prior to labeling. To start radiolabeling, 1.0 mL of dextrose-free media containing 5 μCi/mL $^{14}$C-Acetate was quickly added to each tube, followed by mixing with pipetting. Tubes were tumbled in a 30°C rotating incubator for fifteen minutes. To quench the radiolabeling reaction, and wash the cells, samples were pipetted into 20 mL of −40°C quenching buffer (60% methanol; 1 mM tricine pH 7.4). Cells were incubated in quenching buffer for three minutes, centrifuged at 3000xg at −10°C, and washed with 20 mL of −40°C quenching buffer. Cells were again pelleted by centrifugation, and pellets for lipid extraction were stored at −80°C. For pellets undergoing soluble metabolite extraction, all quenching buffer was thoroughly removed and 1.5 mL of 80°C 75% ethanol was added to each sample followed by a three-minute incubation at 80°C and a subsequent 5-min incubation on ice. Debris was removed from metabolite extract by centrifuging at 20,000xg for 1 min. HMG-CoA labeled during the pulse was separated and quantified as mevalonate. To convert endogenous HMG-CoA to mevalonate, ethanol was thoroughly evaporated from isolated metabolites under argon gas, and each sample was resuspended in HMGCR buffer (50 mM Tris-HCl, pH 6.8; 1 mM NaCl; 1 mM MgCl$_2$; 1 mM DTT; 100 mM glucose-6-phosphate; 1 mM NADP+; 1 mM NADPH). The samples were split into two tubes, one tube would be treated with enzymes, and the untreated tubes acted as blanks for endogenous mevalonate labeled during the pulse. For treated tubes, 2U of HMGCR enzyme (Sigma H7039) and 2U of glucose-6 phosphate dehydrogenase (Sigma G6378) were added. Reactions were carried out over night at 37°C. Prior to loading samples onto TLC plates, total radioactivity in each sample was quantified by scintillation counting and loading was adjusted accordingly. HMGCR Reactions were spotted onto Silica gel G plates (Miles scientific P01911) and separated with a mobile phase of 70:25:5 Diethyl ether:glacial acetic acid:water. Each lane was doped with 5 μg of unlabeled mevalonate to act as a tracer for downstream scraping/quantification. TLC plates were

developed overnight by autoradiography and visualized in an Amersham Typhoon FLA 9500 developer. To visualize the unlabeled tracer mevalonate, plates were sprayed with p-anisaldehyde reagent and baked at 140°C for 10 min. Individual bands were scraped, mixed with 6 mL of EcoLume scintillation cocktail (VWR cat. IC88247001), and quantified by scintillation counting in a Beckman LS 6500 instrument. Mevalonate from the untreated samples was averaged and subtracted from the final values of the treated samples. For lipid extracts, an aliquot of cell lysate was taken immediately following bead beating and radioactivity was quantified by scintillation counting to serve as a normalization standard prior to loading samples onto the TLC plate. Lipid extraction was performed as described above. To separate squalene, ergosterol, DAG, and sterol-esters, total lipid extracts were spotted onto silica gel 60G plates (Fisher cat. NC9825743), and developed in a mobile phase of 55:35:10:1 hexane:petroleum ether:diethyl ether:glacial acetic acid. Autoradiography and scintillation counting was performed as indicated above. Prior to TLC separation, unlabeled squalene, ergosterol, and sterol-ester was added to each lane of the plate as tracer. DAG and ergosterol were quantified by densitometry in Fiji.

## Native protein isolation

Approximately 100OD units of cells were collected for BN-PAGE analysis. Cell pellets were collected into 2 mL screw-cap tubes and frozen at −80°C until further processing. For cell lysis, one large scoop of glass beads and 300 μL of lysis buffer was added to each tube (50 mM HEPES-NaOH, pH 7.0; 50 mM NaCl; 250 mM sorbitol; 10% glycerol; 20 mM arginine; 20 mM glutamic acid; 1.5 mM MgCl$_2$; 1.0 mM CaCl$_2$; 1 mM EDTA; 1 mM DTT; 50 μM Mg132; 1x protease/phosphatase inhibitor cocktail (ThermoFisher cat. 78429)). Tubes containing frozen cell pellet, glass beads and lysis buffer were agitated for one minute using a bead beater followed by 2 min on ice and another minute of agitation with a bead beater. After glass bead lysis, another 700 μL of lysis buffer was added to each tube followed by a brief mixing via vortex. Cell debris was cleared by a 5 min 1000xg spin at 4°C. Supernatants were transferred to a new 1.5 mL tube and centrifuged again at 21,000xg for 30 min at 4°C to isolate a crude microsomal fraction. The supernatants from the first spin were discarded and crude microsomes were washed with 1000 μL of cold lysis buffer followed by another spin at 21,000xg for 30 min at 4°C. Supernatants were again discarded and crude microsomes were resuspended in 60 μL of lysis buffer containing digitonin at a final concentration of 1.5%. Tubes were kept on ice for 30 min and gentle pipetting was applied every 10 min to aid in solubilization of the microsomes. To clear insoluble materials, tubes were centrifuged at 100,000xg for 30 min at 4°C in a Beckman TLA-55 fixed angle rotor. Supernatants were transferred to new tubes on ice after the clearance spin. Approximately 30 μL was reserved for BN-PAGE and 20 μL for SDS-PAGE. The remaining 10 μL was used to quantify protein concentrations in duplicate with the Pierce detergent compatible Bradford kit (ThermoFisher cat. 23246). For SDS-PAGE samples, 4x NuPAGE LDS sample buffer (cat. NP0007) was added to each sample for a final concentration of 1x. Each sample was supplemented with 2-mercaptoethanol and urea at final concentrations of 5% and 8M, respectively. SDS-PAGE samples were heated at 37°C for 40 min and loaded onto homemade 4–15% polyacrylamide gels. SDS-PAGE and immunoblotting was performed as described above.

## Blue native PAGE and immunoblotting

Prior to BN-PAGE, 3.3 μL of BN-PAGE loading buffer was added to each sample (100 mM Bis-Tris, pH 7.0; 500 mM 6-aminocaprioc acid; 5% Coomassie G-250) and gentle mixing was applied via vortex. Samples were loaded onto NativePAGE 3–12% Bis-Tris gels (ThermoFisher cat. BN1001BOX) alongside a NativeMark unstained protein standard (ThermoFisher cat. LC0725). Separation was performed in a mini gel tank (ThermoFisher cat. A25977) with 1x NativePAGE running buffer (ThermoFisher cat. BN2001) in the anode and 1x NativePAGE running buffer containing 1x cathode buffer additive (ThermoFisher cat. BN2002) in the cathode. Electrophoresis was performed at 150V for 30 min, after which cathode buffer was replaced with 1x NativePAGE running buffer containing 0.1x cathode buffer additive and ran at 150V for another 30 min. After 1 hr at 150V, voltage was increased to 300V until the dye front was at the bottom of the gel. Once gel case was removed, the wells and dye front were excised and discarded from the gel with a clean razor blade. The ladder was excised and stained separately in GelCode blue stain (ThermoFisher cat. 24594) according to the manufacturer's instructions. The remaining gel was incubated in denaturing buffer (300 mM Tris;

100 mM acetic acid; 1% SDS) for 20 min on a rocker, after which, the gel was placed between two glass plates wetted with denaturing buffer and incubated for one hour. Proteins were transferred to a 0.2 μm PVDF membrane overnight by wet transfer at 25V in a cold room with BN-PAGE transfer buffer (150 mM Tris; 50 mM acetic acid). After overnight transfer, the membrane was incubated in 8% acetic acid for five minutes followed by a 5-min incubation in MiilliQ water. The membrane was dried for 1 hr at room temperature and then washed quickly four times with 100% methanol followed by a quick wash with MilliQ water. The membrane was then placed in 5% milk for 1 hr, and subsequent steps for immunoblotting were performed as described above. For detection of Hmg1-3HA, an anti-HA primary antibody was used (Abcam ab9110; 1:2000 dilution) followed by rabbit secondary described above.

## Statistical analysis

T-tests and one-way ANOVA tests were performed using Graphpad Prism8 software. Kolmogorov-Smirnov test was conducted using kstest2 MATLAB function. All unpaired t-tests were performed with Welch's correction. For one-way ANOVA, Brown-Forsyth and Welch ANOVA was performed to account for non-uniform standard deviations, followed by Turkey's post-hoc test to extract p-values. For all t-tests and ANOVAs. * $p$-value<0.05, **$p$-value<0.01, ***$p$-value<0.001. Dotted lines in violin plots represent the median of the data, while upper and lower dotted lines represent the upper and lower quartiles. Bar graphs show mean of the data with error bars indicating the standard deviation.

## Acknowledgements

The authors want to thank members of the Henne and Nicastro labs for help in the completion of this project. The authors thank Joel Goodman for the *tgl3,4,5,Δ* yeast strain. W.M.H. is supported by funds from the Welch Foundation (I-1873), the NIH NIGMS (GM119768), the Ara Parseghian Medical Research Fund, and the UT Southwestern Endowed Scholars Program. S.R. is supported in part by a NIH T32 training grant (5T32GM008297).

## Additional information

### Funding

| Funder | Grant reference number | Author |
|---|---|---|
| Welch Foundation | I-1873 | W Mike Henne |
| National Institute of General Medical Sciences | GM119768 | W Mike Henne |
| National Institute of General Medical Sciences | 5T32GM008297 | Sean Rogers |
| National Institute of Diabetes and Digestive and Kidney Diseases | DK126887 | W Mike Henne |

The funders had no role in study design, data collection and interpretation, or the decision to submit the work for publication.

### Author contributions

Sean Rogers, Conceptualization, Data curation, Formal analysis, Visualization, Methodology, Writing - original draft; Hanaa Hariri, Conceptualization, Data curation, Supervision; N Ezgi Wood, Conceptualization, Data curation, Formal analysis, Visualization, Methodology; Natalie Ortiz Speer, Conceptualization, Visualization, Methodology; W Mike Henne, Conceptualization, Formal analysis, Supervision, Funding acquisition, Investigation, Writing - original draft, Project administration

### Author ORCIDs

Sean Rogers (iD) https://orcid.org/0000-0003-1173-6526
Hanaa Hariri (iD) https://orcid.org/0000-0002-2953-7536

N Ezgi Wood (iD) https://orcid.org/0000-0002-1874-9226
Natalie Ortiz Speer (iD) http://orcid.org/0000-0002-0982-3025
W Mike Henne (iD) https://orcid.org/0000-0002-2135-2799

**Decision letter and Author response**
Decision letter https://doi.org/10.7554/eLife.62591.sa1
Author response https://doi.org/10.7554/eLife.62591.sa2

## Additional files

### Supplementary files

- Supplementary file 1. Table S1: Yeast strains used in this study.
- Supplementary file 2. Table S2: Plasmids used in this study.
- Transparent reporting form

### Data availability

All data generated or analysed during this study are included in the manuscript and supporting files.

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
