## [Decision Letter]

**Acceptance summary:**

This study shows that an important biosynthetic pathway is influenced by the sequestration of some of its enzymes at a membrane contact site. This finding uncovers a novel way to control metabolism, besides transcriptional, translational and posttranslational control; the controlled doing and undoing of contact sites.

**Decision letter after peer review:**

Thank you for submitting your article "Glucose restriction drives spatial reorganization of mevalonate metabolism and liquid-crystalline lipid droplet biogenesis" for consideration by *eLife*. Your article has been reviewed by three peer reviewers, including Benoît Kornmann as the Reviewing Editor and Reviewer #1, and the evaluation has been overseen by Suzanne Pfeffer as the Senior Editor.

The reviewers have discussed the reviews with one another and the Reviewing Editor has drafted this decision to help you prepare a revised submission.

This study shows that the yeast HMG-CoA reductase, which normally localizes throughout the ER, is, upon acute glucose removal (AGR), sequestered at a contact site between the nuclear ER and the lysosome, called nuclear-vacuole junctions (NVJ). AGR is accompanied by an increase in sterol ester synthesis, and failure to generate NVJs hampers the flux in the steryl ester biosynthesis pathway, suggesting that the sequestration of the HMG-CoA reductase impacts its activity positively. One of the strengths of the manuscript is that it takes a great care in removing any transcriptional influence by observing changes in HMG-CoA reductase activity when the protein is expressed from constitutive promoters, and also that it attempts to recreate the activity change by artificially clustering of the protein at defined positions. This novel finding stresses the importance of subcellular localization for proper wiring of metabolic fluxes. Additionally, AGR causes the appearance of liquid-crystalline structures in lipid droplets, an interesting side observation.

All three reviewers found the work exciting and important. There are nevertheless ambiguities in the present manuscript that need to be addressed, and a decision taken about what to make of the side observation with liquid crystalline structures in LDs (see below).

Points that require new experimental work:

1. The first point is about the artificial clustering. This is a very important point of the study, but the result is however yet not entirely clear.

– The human HMG-CoA reductase works as a tetramer. The oligomerization state of yeast Hmg1 is unclear. There is a strong focus on the manuscript on multimerization without data supporting it. Monomeric mRuby and tetrameric DsRed are used to manipulate the oligomeric state Hmg1 but there is no information on the oligomeric state of these fusion proteins. Moreover it is not tested whether the fusion to different tags affect the protein's activity and/or abundance. In order to make the point that it is indeed the oligomerization status change that causes the change in activity, it is necessary to rule out changes in abundance or activity due to the tags being appended to the protein.

– It is unclear why AGR should cause stabilized Hmg1-dsRed2 multimers in nvj1-delete cells (lines 243-244, figure 6I). Here the multimerization is supposed to be constitutive and driven by the dsRed. The factor (Nvj1) that is supposed to cause AGR-dependent clustering of Hmg1 is absent. Does it mean that Hmg1 has a tendency to cluster during AGR in a way that is mechanistically unclear, and in order for it to happen efficiently, it needs to be boosted, either by being sequestered at the NVJ, or by being fused to a tetramerizing tag? This definitely requires further tests and explanations.

2. The second point is Nvj1's influence on Hmg1.

– One likely explanation that is strongly implied is that Nvj1 and Hmg1 interact via the 10 a.a. stretch identified therein. Can this interaction be shown by pull-down? Is the interaction induced by AGR, or is it constitutive (in which case Hmg1 simply follows Nvj1 relocalisation to NVJs in AGR)? Also several conditions prevent Hmg1 recruitment and sequestration to NVJs (e.g. lovastatin, upc2 deletion). Do these conditions affect the making of NVJs and the localisation of Nvj1 therein or are they specific to Hmg1 recruitment?

– There is a possibility that the effects of the Nvj1 deletions are indirect through the absence of NVJ, and the absence of recruitment therein of other lipid homeostasis factors (e.g. Osh1, Tsc13, Vps13). It is however possible to decouple NVJ generation and Hmg1 recruitment by using the 10 a.a. deletion mutant. Using such a mutant would be much cleaner and ideally, all experiments using Nvj1 delete cells should be replaced or complemented with the 10 a.a. mutant. This would imply a large amount of work, and the reviewers discussed that it should be sufficient to just do the growth assays, as well as the data in figure 5A-G with this setup (see also point below).

– It is emphasized throughout that Nvj1's influence on Hmg1 activity is limited to AGR conditions. This is consistent with Hmg1 sequestrations at NVJs only in these conditions. It is however not tested.

All the data from figure 5C-G and 6D-G,J-M are in AGR only. Yet, Nvj1 might also influence the mevalonate pathway in log phase. Since it will be anyway necessary to redo the experiments of figure 5A-G with the 10 a.a. nvj1 mutant (see point above), it will be necessary to add log phase quantification. If nvj1 mutation or deletion had an effect on log phase, a condition where Hmg1 is not clustered at NVJs, then this would imply a much more indirect mechanism. To make the claim water-tight, these experiments should ideally be performed in an Hmg2-delete background to discount any variations in Hmg2 expression during AGR.

Points that might be addressed by amendments in the text, or by new experimental work:

3. The part on Liquid crystalline layered LDs is disconnected from the rest. It is not tested whether Nvj1-dependent clustering of Hmg1 to NVJs is necessary, sufficient or unrelated to the formation of these structure. While all three reviewers found it to be an interesting observation, they found that, as presently written, the manuscript is ambiguous at stating the lack of connection between the two phenomena. The title itself "Glucose restriction drives spatial re-organization of mevalonate metabolism and liquid-crystalline lipid droplet biogenesis" could suggest that the latter is the result of the former, which is not tested. It would be crucial to either strongly emphasize the absence of connection between the two stories (including likely remove reference to the LCL^-^LDs in the title), find a link between the stories (e.g. show that LCL^-^LDs are not made with the 10 a.a. mutant), or remove this part entirely to keep it for another, more focused manuscript.

4. The effect of Upc2 on Hmg1 is unclear. The paragraph starts with the observation of increased Hmg1 accumulation in AGR, then looks for factor affecting this induction and homes in on Upc2. There, however, instead of protein accumulation, the subcellular localization is assessed. Does Upc2 deletion blunt Hmg1 induction in AGR or does it only affect its localization? If the latter, is it because of a general imbalance in the sterol biosynthesis pathway, akin to lovastatin-treated cells? Does it also affect Njv1 localization to the vacuole, or does it affect Hmg1-Nvj1 interaction (if not constitutive, see above)? While deciphering the exact mechanistic role of Upc2 in this context is probably beyond the scope of this manuscript, it should however be discussed clearly in the text.

5. Seo and colleagues (*eLife* 2016) showed that AGR triggers LD autophagy. While the two phenotypes observed are not incompatible, the apparent differences of AGR in LD formation and/or consumption should at least be discussed.

6. The reduced ability of nvj1 deleted cells to cope with AGR is attributed to reduced sterylester synthesis. Since Nvj1-delete cells might be suffering from several other ailments, a more direct way to test the importance of sterylester biosynthesis in AGR survival could be to look at are1,2 deleted cells.

Please also have a careful look at and address the points of the individual reviews appended below.

Reviewer #1:

This study reports that acute glucose restriction (AGR) causes the sequestration of HMG-Coa reductases at Nucleus-Vacuole Junctions (NVJs), contact sites between the nuclear ER and lysosomal membranes in yeast. This sequestration is shown to enhance the metabolic activity of the enzyme, something that is beneficial to survive AGR. In a second part, AGR is shown to cause phase separation of steryl-esters in lipid droplets, but the connection to NVJ and HM-Coa reductase sequestration is unclear. The authors hammer their points home with elegant genetics where they artificially cluster HMG-Coa reductase and remove any transcriptional influence.

This is a very interesting story that shows that mere subcellular repositioning of enzymes can affect their activity and change metabolic fluxes. It is one of the first study that pinpoints a role for membrane contact sites beyond the classical lipid and calcium exchanges.

Below are some ideas to strengthen the main claims of the paper:

1. Throughout, the importance of HMG-CoA reductase sequestration is assessed by comparing wild-type strains to a NVJ1-delete strain, which is incapable of sequestering HMG-CoA reductase to NVJs. However the deletion of Nvj1 affects much more than HMG-CoA sequestration. It affects the whole making of NVJs and the sequestration therein of several factors which could have indirect roles in the phenotypes observed, for instance Osh1, Vps13 and Tsc13. Incidentally, the authors present a much cleaner way to assess the role of HMG-CoA reductase sequestration; a mutant lacking just 5 amino acids from Nvj1, which is still able to make NVJs and presumably still able to recruit Osh1, Vps13 and Tsc13 therein. It would be much more informative if all experiments using NVJ1-delete cells were performed while rescuing the phenotype with either WT Nvj1 or the 5-aminoacid deletion thereof. This would tease apart what from the phenotype is due to a lack in HMG CoA reductase sequestration and what is due to lack of NVJs in general.

2. There is a confusion regarding Upc2's activity in regulating Hmg1. It is proposed that this transcription factor is responsible for the increased accumulation of Hmg1 in AGR. Yet, the test for this idea, does not look at Hmg1 accumulation, but at its distribution. Accumulation and distribution are two separate processes that can be teased apart as shown in figure 6A-C. It appears, from figure 4E that Upc2 rather affects the distribution of Hmg1. We do not know if it affects its accumulation, since a western blot (or any other quantification) has not been performed. Even if Upc2 was necessary for transcriptional induction of Hmg1, a change in distribution of the preexisting pool of Hmg1 should be observed, unless only the newly synthesized pool is susceptible to sequestration. This should be tested, and is not quite consistent with the experiments where Hmg1 is under the control of the ADH1 promoter. An alternate hypothesis is that Upc2 is required for the expression of another factor that is necessary for Hmg1 sequestration. This hypothesis appears quite likely and can be tested by deleting UPC2 in the strain where Hmg1 is under the Adh1 promoter.

3. The experiments with artificial clustering are only valid if the different fusion proteins are expressed at the same levels. This is not tested.

Reviewer #2:

In this manuscript, Rogers et al. show how in response to acute glucose restriction, HMG-CoA Reductases re-localize from the nuclear envelope to the nucleus vacuole junction (NVJ), a contact site at which the nucleus and vacuole are physically attached to each other. They describe in detail the molecular determinants underlying this behavior, and uncover a functional role of spatial enzyme compartmentalization in regulation of flux through the mevalonate pathway, and a link to formation of liquid-crystalline layered sterol ester-rich lipid droplets.

These are conceptually novel findings that will likely be of high interest to the broad readership of *eLife*. The data is of very high quality, and supports the claims raised by the authors.

I support publication of this exciting manuscript.

Reviewer #3:

This manuscript describes that during acute glucose restriction (AGR) the rate limiting enzyme of the mevalonate pathway Hmg1 concentrates at the NVJ. Hmg1 relocalization requires the NVJ components Nvj1, Vac8, the ergosterol-sensing transcription factor Upc2 and the activity of the sterol enzyme Erg1. Hmg1 partitioning is mediated through its transmembrane domain and requires a 10 amino acid region in Nvj1 luminal domain. It is claimed that Hmg1 NVJ partitioning promotes increased flux through the mevalonate pathway and accumulation of steryl ester to promote growth during AGR recovery. Some of the observation described are very interesting. Moreover, understanding how metabolic pathways and membrane contacts sites are rewired in response to environmental changes is an issue of fundamental importance. However, the study presents a number of critical flaws and some of the claims are not convincingly demonstrated.

1. A major assumption of this study is that Hmg1 does not normally function as a tetramer as mammalian HMGR. In mammals HMGR tetramerization is mediated by its cytosolic catalytic domain which is more than 60% similar to Hmg1. This complicates the interpretation of several experiments. For example, the authors the authors use monomeric mRuby and tetrameric DsRed to "manipulate" the oligomeric state Hmg1 but there is really no information on the oligomeric state of these fusion proteins. The results presented in Figure 6 can easily be explained by differences in the levels or functionality of the two fusion proteins (not tested in both cases).

2. It is nicely shown that Hmg1 NVJ localization requires its membrane domain and a few residues in the luminal domain of Nvj1. But how exactly is this regulated? Do these proteins interact? If yes, is this interaction constitutive and driven by the increase in Nvj1 upon AGR? or is it regulated in response to AGR?

3. Still related with the previous point, the trigger for Hmg1 NVJ localization is unclear. The upc2D and terbinafine results suggest the flux through the pathway is important but some additional information would be good. There are drugs/mutants affecting almost all the steps along the pathway so this could be easily resolved.

4. In figure 5A-B, it is shown that inhibition of Hmg1 with Lovastatin reverses the changes in steryl esters as expected. Does this treatment also affect Hmg1 NVJ accumulation and/or Nvj1 distribution?

5. All the measurements of mevalonate pathway intermediates and products were performed in cells expressing Hmg2 and the levels of the protein under the conditions tested is unknown. Is it possible that Nvj1 affects the levels of Hmg2? To avoid this complications, the same measurements should have been done in hmg2D cells. Also the data shown in Figure 5 and 6 does not show that the effect is specific to AGR

6. The authors propose that steryl esters are necessary for optimal AGR response. However, this possibility was not formally tested by preventing steryl ester formation, which can be easily done by deleting Are1 and Are2 acyl transferases.

7. In Figure 6I, why does Hmg1-DsRed2 responds to AGR even in nvj1D cells. Presumably the tetramerization of DsRed is independent of glucose levels.

8. The data on LCL^-^LDs is very interesting but it does not seem to belong to this story.

9. Seo and colleagues (*eLife* 2016) showed that AGR triggers LD autophagy. While the two phenotypes observed are not incompatible, the apparent differences of AGR in LD formation and/or consumption should at least be discussed.

---

## [Author Response]

All three reviewers found the work exciting and important. There are nevertheless ambiguities in the present manuscript that need to be addressed, and a decision taken about what to make of the side observation with liquid crystalline structures in LDs (see below).Points that require new experimental work:1. The first point is about the artificial clustering. This is a very important point of the study, but the result is however yet not entirely clear.– The human HMG-CoA reductase works as a tetramer. The oligomerization state of yeast Hmg1 is unclear. There is a strong focus on the manuscript on multimerization without data supporting it. Monomeric mRuby and tetrameric DsRed are used to manipulate the oligomeric state Hmg1 but there is no information on the oligomeric state of these fusion proteins. Moreover it is not tested whether the fusion to different tags affect the protein's activity and/or abundance. In order to make the point that it is indeed the oligomerization status change that causes the change in activity, it is necessary to rule out changes in abundance or activity due to the tags being appended to the protein.

We thank the reviewers for this excellent point. We have now analyzed both Hmg1-mRuby3 and Hmg1-DsRed2 protein expression in WT and nvj1-KO backgrounds by Western blot, and found very similar trends for Hmg1-GFP levels in WT and nvj1-KO backgrounds. That is to say, Hmg1 expression increases in AGR very similarly in both strains, suggesting that our radio-pulse data cannot be simply explained by differential protein expression. Unfortunately, due to lack of antibodies recognizing untagged Hmg1, we cannot compare Hmg1 protein expression directly *between* Hmg1-mRuby3 and Hmg1-DsRed2 strains. We attempted several ways to do this, including use of multiple antibodies raised against human HMGCR, but these would not recognize yeast Hmg1. We also contacted other members of the yeast community that have studied Hmg1 in the past, but no antibodies were available.

Secondly, to address the point about Hmg1 oligomerization: (1) We now directly observe Hmg1 high molecular weight (HMW) species by blue native PAGE (Figure 7A,B). Importantly, these HMW species are induced by AGR, and are Nvj1-dependent. This supports a model where AGR induces Hmg1 to partition at the NVJ, which promotes Hmg1 incorporation and/or assembly into a HMW species. (2) To be as precise on our interpretation as possible, we have changed the phrasing used in this paper to describe Hmg1 entering into ‘High Molecular Weight Species’ rather than explicitly stating that these are Hmg1 tetramers.

Third, regarding the use of DsRed2 to modulate Hmg1 multimerization: we attempted several different methods to directly monitor Hmg1-DsRed2 oligomeric status. Indeed, by BN gels, we observed that Hmg1-DsRed2 ran at high molecular weights, but these gels were often of lower quality as the Hmg1-DsRed2 bands were quite smeary. We also noted that Hmg1-DsRed2 ran at a higher than expected molecular weight even in SDS-PAGE gels, indicating that this tag promoted SDS-resistant multimers. We have now added a comparative SDS-PAGE gel showing this and comparing it to Hmg1-mRuby3 in Supp Figure 5E. You can also note the smear-like pattern of the Hmg1-DsRed2 in Supp Fig5D.

Fourth, regarding any changes to Hmg1 abundance in different genetic backgrounds: We noted above that Hmg1-DsRed2 and Hmg1-mRuby3 both display comparative increases in protein abundance with AGR, suggesting these tags to not alter the AGR-induced protein increases for Hmg1. Additionally, because we now have an Nvj1 mutant (Nvj1 RK-28,29-AA) that exhibits mevalonate pathway defects, we quantified Hmg1-mRuby3 fluorescence intensities when expressed in Nvj1 WT and Nvj1 RK-28,29-AA backgrounds. The Hmg1-mRuby3 fluorescence intensity distributions are near identical in both backgrounds, suggesting the alterations to mevalonate pathway flux in the Nvj1 RK-28,29-AA background are not due to differences in Hmg1 protein levels (Figure 6—figure supplement 1C).

– It is unclear why AGR should cause stabilized Hmg1-dsRed2 multimers in nvj1-delete cells (lines 243-244, figure 6I). Here the multimerization is supposed to be constitutive and driven by the dsRed. The factor (Nvj1) that is supposed to cause AGR-dependent clustering of Hmg1 is absent. Does it mean that Hmg1 has a tendency to cluster during AGR in a way that is mechanistically unclear, and in order for it to happen efficiently, it needs to be boosted, either by being sequestered at the NVJ, or by being fused to a tetramerizing tag? This definitely requires further tests and explanations.

Thank you for pointing this out. We agree: our point here was that the DsRed2 tag is designed to generate constitutive multimers for Hmg1. Indeed, we observe that this DsRed2 tag promotes Hmg1-DsRed2 to run at high molecular weight even in SDS-PAGE gels in Log or AGR conditions (Figure 7—figure supplement 1E). We do see more Hmg1-DsRed2 foci by imaging during AGR, but this may be due to the increase in total Hmg1-DsRed2 protein induced by AGR, which is similar to other tags (Figure 7—figure supplement Supp 1D). We have amended the text to try to articulate this as precisely as possible. Our current model for how Hmg1 multimerizes, which we now state in the Discussion, is that NVJ compartmentalization of Hmg1/2 may promote the probability of these enzymes, which normally diffuse throughout the ER network, to engage each other and multimerize. In essence, the NVJ may serve as an ER sub-domain where enzymes may have increased probability to engage one another.

2. The second point is Nvj1's influence on Hmg1.– One likely explanation that is strongly implied is that Nvj1 and Hmg1 interact via the 10 a.a. stretch identified therein. Can this interaction be shown by pull-down? Is the interaction induced by AGR, or is it constitutive (in which case Hmg1 simply follows Nvj1 relocalisation to NVJs in AGR)? Also several conditions prevent Hmg1 recruitment and sequestration to NVJs (e.g. lovastatin, upc2 deletion). Do these conditions affect the making of NVJs and the localisation of Nvj1 therein or are they specific to Hmg1 recruitment?

Thank you for this suggestion. Working with Hmg1 in biochemical experiments has proven very challenging as Hmg1 is prone to aggregation. Thus, we were concerned with co-IP data. However, we now present new data showing that mutation of Nvj1 residues RK-28,29 to alanine (RK— 28,29—AA) inhibits Hmg1 recruitment to the NVJ while maintaining NVJ patch formation (Figure 3G-I). This Nvj1 mutant enabled us to show that loss of Hmg1 NVJ recruitment perturbs mevalonate pathway flux (Figure 6A-D), and growth resumption after glucose starvation (Figure 6E,F) even while the NVJ is intact. We also now show that loss of Upc2 does not affect NVJ formation (Figure 4—figure supplement 1A-C). We did not use lovastatin for these experiments. We also now comment in the Discussion that we cannot rule out the fact that the Nvj1-Hmg1 interaction may be indirect via another protein, but fully elucidating this interaction is beyond the scope of the study.

– There is a possibility that the effects of the Nvj1 deletions are indirect through the absence of NVJ, and the absence of recruitment therein of other lipid homeostasis factors (e.g. Osh1, Tsc13, Vps13). It is however possible to decouple NVJ generation and Hmg1 recruitment by using the 10 a.a. deletion mutant. Using such a mutant would be much cleaner and ideally, all experiments using Nvj1 delete cells should be replaced or complemented with the 10 a.a. mutant. This would imply a large amount of work, and the reviewers discussed that it should be sufficient to just do the growth assays, as well as the data in figure 5A-G with this setup (see also point below).

As mentioned above, we now show that a mutant Nvj1 (Nvj1 RK-28,29-AA) shows very similar defects in mevalonate pathway flux AND growth resumption in a background where the NVJ is still intact (Figure 6A-F). This strongly suggests that the effects we observe are due to Hmg1 not being recruited to the NVJ, and not from the general loss of the NVJ itself.

– It is emphasized throughout that Nvj1's influence on Hmg1 activity is limited to AGR conditions. This is consistent with Hmg1 sequestrations at NVJs only in these conditions. It is however not tested.All the data from figure 5C-G and 6D-G,J-M are in AGR only. Yet, Nvj1 might also influence the mevalonate pathway in log phase. Since it will be anyway necessary to redo the experiments of figure 5A-G with the 10 a.a. nvj1 mutant (see point above), it will be necessary to add log phase quantification. If nvj1 mutation or deletion had an effect on log phase, a condition where Hmg1 is not clustered at NVJs, then this would imply a much more indirect mechanism. To make the claim water-tight, these experiments should ideally be performed in an Hmg2-delete background to discount any variations in Hmg2 expression during AGR.

We attempted to monitor mevalonate pathway flux during Log phase via radiolabel pulse-chase experiments. Surprisingly, the uptake of ^14^C-acetate is very low in Log phase. Yeast in Log phase growth thus exhibit very low ^14^C-acetate incorporation into mevalonate pathway products. For comparison, we’ve attached a cpm histogram for yeast in Log and AGR phases in Author response image 1. This low ^14^C-acetate uptake made monitoring 14-C labeled mevalonate pathway products essentially impossible.

Regarding whether nvj1 mutation or deletion has an effect in Log phase, it should be noted that the Nvj1 RK-28,29-AA mutant experiments in AGR demonstrate that defects in mevalonate pathway flux still occur when the NVJ is intact. This also addresses the indirect effect issue.Regarding variation from Hmg2 expression, we have conducted experiments in our artificial strain that only expresses Hmg1-GFP on an artificial promoter (Figure 6H-M), as well as a strain background expressing only Hmg1 (Hmg2 also deleted here) with a mutant Nvj1 that cannot recruit Hmg1 (Figure 6A-D). Both experiment sets indicate that loss of Hmg1 recruitment to the NVJ perturbs mevalonate pathway flux.

Points that might be addressed by amendments in the text, or by new experimental work:3. The part on Liquid crystalline layered LDs is disconnected from the rest. It is not tested whether Nvj1-dependent clustering of Hmg1 to NVJs is necessary, sufficient or unrelated to the formation of these structure. While all three reviewers found it to be an interesting observation, they found that, as presently written, the manuscript is ambiguous at stating the lack of connection between the two phenomena. The title itself "Glucose restriction drives spatial re-organization of mevalonate metabolism and liquid-crystalline lipid droplet biogenesis" could suggest that the latter is the result of the former, which is not tested. It would be crucial to either strongly emphasize the absence of connection between the two stories (including likely remove reference to the LCL^-^LDs in the title), find a link between the stories (e.g. show that LCL^-^LDs are not made with the 10 a.a. mutant), or remove this part entirely to keep it for another, more focused manuscript.

We thank the reviewers for these constructive comments. We have now removed the data related to the liquid-crystalline LDs in this revision. The revision is thus focused on dissecting the role of Hmg1 compartmentalization on mevalonate pathway flux during AGR.

4. The effect of Upc2 on Hmg1 is unclear. The paragraph starts with the observation of increased Hmg1 accumulation in AGR, then looks for factor affecting this induction and homes in on Upc2. There, however, instead of protein accumulation, the subcellular localization is assessed. Does Upc2 deletion blunt Hmg1 induction in AGR or does it only affect its localization? If the latter, is it because of a general imbalance in the sterol biosynthesis pathway, akin to lovastatin-treated cells? Does it also affect Njv1 localization to the vacuole, or does it affect Hmg1-Nvj1 interaction (if not constitutive, see above)? While deciphering the exact mechanistic role of Upc2 in this context is probably beyond the scope of this manuscript, it should however be discussed clearly in the text.

We now show that Upc2 deletion does not alter the behavior of Hmg1 protein levels (Figure 4figure supplement 1D). We also now show that Upc2-ko does not impact NVJ formation or Nvj1 levels (Figure 4—figure supplement 1A-C). We agree that completely dissecting the interplay between Upc2, Nvj1, and Hmg1 is beyond the scope of this manuscript, but have revised the text to say that the full mechanism will be explored in future studies.

5. Seo and colleagues (eLife 2016) showed that AGR triggers LD autophagy. While the two phenotypes observed are not incompatible, the apparent differences of AGR in LD formation and/or consumption should at least be discussed.

Thank you for this important discussion point. We now discuss the relationship between glucosestarvation induced autophagy and glucose restriction in the Discussion. We speculate that HMGCR partitioning at the NVJ may impact aspects of LD biogenesis or composition, which may ultimately impact the micro-autophagy of LDs. These are a part of ongoing studies.

6. The reduced ability of nvj1 deleted cells to cope with AGR is attributed to reduced sterylester synthesis. Since Nvj1-delete cells might be suffering from several other ailments, a more direct way to test the importance of sterylester biosynthesis in AGR survival could be to look at are1,2 deleted cells.

We find that the growth resumption defects associated with nvj1-ko can be rescued by culturing the yeast with mevalonate, the enzymatic product of HMGCRs. Therefore, we attribute the defects with perturbed mevalonate synthesis itself, rather than sterol-ester synthesis, per se. Mevalonate is a key metabolite needed for several metabolic pathways including sterol metabolism but also coenzyme Q and heme production, which are needed for mitochondrial metabolism. We speculate that growth resumption defects may be related to mitochondrial metabolism, but fully resolving this is beyond the scope of this current study.

Please also have a careful look at and address the points of the individual reviews appended below.Reviewer #1:This study reports that acute glucose restriction (AGR) causes the sequestration of HMG-Coa reductases at Nucleus-Vacuole Junctions (NVJs), contact sites between the nuclear ER and lysosomal membranes in yeast. This sequestration is shown to enhance the metabolic activity of the enzyme, something that is beneficial to survive AGR. In a second part, AGR is shown to cause phase separation of steryl-esters in lipid droplets, but the connection to NVJ and HM-Coa reductase sequestration is unclear. The authors hammer their points home with elegant genetics where they artificially cluster HMG-Coa reductase and remove any transcriptional influence.This is a very interesting story that shows that mere subcellular repositioning of enzymes can affect their activity and change metabolic fluxes. It is one of the first study that pinpoints a role for membrane contact sites beyond the classical lipid and calcium exchanges.Below are some ideas to strengthen the main claims of the paper:1. Throughout, the importance of HMG-CoA reductase sequestration is assessed by comparing wild-type strains to a NVJ1-delete strain, which is incapable of sequestering HMG-CoA reductase to NVJs. However the deletion of Nvj1 affects much more than HMG-CoA sequestration. It affects the whole making of NVJs and the sequestration therein of several factors which could have indirect roles in the phenotypes observed, for instance Osh1, Vps13 and Tsc13. Incidentally, the authors present a much cleaner way to assess the role of HMG-CoA reductase sequestration; a mutant lacking just 5 amino acids from Nvj1, which is still able to make NVJs and presumably still able to recruit Osh1, Vps13 and Tsc13 therein. It would be much more informative if all experiments using NVJ1-delete cells were performed while rescuing the phenotype with either WT Nvj1 or the 5-aminoacid deletion thereof. This would tease apart what from the phenotype is due to a lack in HMG CoA reductase sequestration and what is due to lack of NVJs in general.

Thank you. As discussed above, we now show that Nvj1 mutant RK-28,29-AA, which can form an NVJ but does not recruit Hmg1, manifests defects in mevalonate pathway flux (Figure 6A-D) and growth resumption following glucose starvation (Figure 6E,F). This indicates that these effects are specific for Hmg1 recruitment, and not due to general loss of the NVJ.

2. There is a confusion regarding Upc2's activity in regulating Hmg1. It is proposed that this transcription factor is responsible for the increased accumulation of Hmg1 in AGR. Yet, the test for this idea, does not look at Hmg1 accumulation, but at its distribution. Accumulation and distribution are two separate processes that can be teased apart as shown in figure 6A-C. It appears, from figure 4E that Upc2 rather affects the distribution of Hmg1. We do not know if it affects its accumulation, since a western blot (or any other quantification) has not been performed. Even if Upc2 was necessary for transcriptional induction of Hmg1, a change in distribution of the preexisting pool of Hmg1 should be observed, unless only the newly synthesized pool is susceptible to sequestration. This should be tested, and is not quite consistent with the experiments where Hmg1 is under the control of the ADH1 promoter. An alternate hypothesis is that Upc2 is required for the expression of another factor that is necessary for Hmg1 sequestration. This hypothesis appears quite likely and can be tested by deleting UPC2 in the strain where Hmg1 is under the Adh1 promoter.

We now show that upc2-ko does not effect Hmg1 protein levels (Figure 4—figure supplement 1D). We also now show that upc2-ko does not alter NVJ formation nor Nvj1 levels (Figure 4—figure supplement 1A-C). Regarding other factors Upc2 may regulate the expression of, we agree, and have modified the text to state that the full mechanism of Upc2’s role in the Nvj1-Hmg1 interaction requires further study beyond the scope of this study.

3. The experiments with artificial clustering are only valid if the different fusion proteins are expressed at the same levels. This is not tested.

We now provide evidence that the Hmg1 protein levels undergo the same changes in abundance during Log and AGR, suggesting these differences are not due to changes in abundance. Furthermore, our new experiments with Nvj1 RK-28,29-AA show that loss of Hmg1 recruitment to the NVJ causes defects in mevalonate pathway flux. These yeast contain the same Hmg1-mRuby3 protein levels (Figure 6—figure supplement 1C), suggesting that alterations in Hmg1 partitioning, and not protein level, cause defects im mevaloante pathway flux.

Reviewer #3:This manuscript describes that during acute glucose restriction (AGR) the rate limiting enzyme of the mevalonate pathway Hmg1 concentrates at the NVJ. Hmg1 relocalization requires the NVJ components Nvj1, Vac8, the ergosterol-sensing transcription factor Upc2 and the activity of the sterol enzyme Erg1. Hmg1 partitioning is mediated through its transmembrane domain and requires a 10 amino acid region in Nvj1 luminal domain. It is claimed that Hmg1 NVJ partitioning promotes increased flux through the mevalonate pathway and accumulation of steryl ester to promote growth during AGR recovery. Some of the observation described are very interesting. Moreover, understanding how metabolic pathways and membrane contacts sites are rewired in response to environmental changes is an issue of fundamental importance. However, the study presents a number of critical flaws and some of the claims are not convincingly demonstrated.1. A major assumption of this study is that Hmg1 does not normally function as a tetramer as mammalian HMGR. In mammals HMGR tetramerization is mediated by its cytosolic catalytic domain which is more than 60% similar to Hmg1. This complicates the interpretation of several experiments. For example, the authors the authors use monomeric mRuby and tetrameric DsRed to "manipulate" the oligomeric state Hmg1 but there is really no information on the oligomeric state of these fusion proteins. The results presented in Figure 6 can easily be explained by differences in the levels or functionality of the two fusion proteins (not tested in both cases).

Thank you for this important point. First, as discussed above, we now present data that Hmg1 enters into high molecular weight (HMW) assemblies that we observe by Blue Native Gel (Figure 7A,B). These HMW assemblies are induced by AGR, and are Nvj1 dependent. Second, we have now amended our text to say that Hmg1 “enters into high molecular weight assemblies” rather than make any claims that Hmg1 forms tetramers. Third, we have conducted new Western blots showing that Hmg1-mRuby3 and Hmg1-DsRed2 tagged proteins undergo very similar protein increased from Log to AGR conditions, suggesting tagging does not affect overall protein levels. Fourth, we now provide evidence that Hmg1-DsRed2 runs at higher than predicted MWs even in SDS-PAGE gels, indicating the DsRed2 tetramer promotes oligomerization (Figure 7—figure supplement 1E). Fifth, we also now show that in a yeast background where the Hmg1 protein levels are the same (Figure 6—figure supplement 1C), when Hmg1 cannot be recruited to the NVJ (as in Nvj1 RK-28,29-AA), there are defects in mevalonate pathway flux (Figure 6A-D).

2. It is nicely shown that Hmg1 NVJ localization requires its membrane domain and a few residues in the luminal domain of Nvj1. But how exactly is this regulated? Do these proteins interact? If yes, is this interaction constitutive and driven by the increase in Nvj1 upon AGR? or is it regulated in response to AGR?

As discussed above, we now present data that this Nvj1-Hmg1 interaction requires residues RK28,29, and use a double alanine mutant to show that this mutant cannot recruit Hmg1, has defects in mevalonte metabolism flux, but still forms an NVJ (Figure 6A-D, Figure 3H,I). We have also modified the text to state that we do not fully understand this mechanism of Upc2-Nvj1-Hmg1 interaction, but this is part of an ongoing investigation beyond the scope of this manuscript.

3. Still related with the previous point, the trigger for Hmg1 NVJ localization is unclear. The upc2D and terbinafine results suggest the flux through the pathway is important but some additional information would be good. There are drugs/mutants affecting almost all the steps along the pathway so this could be easily resolved.

We now show that upc2-ko does not affect Hmg1 protein levels (Figure 6—figure supplement 1D). We also more further defined the interaction as requiring Nvj1 residues RK-28,29. As mentioned above, we do not fully understand the mechanism of this interaction, but it is beyond the scope of this current paper.

4. In figure 5A-B, it is shown that inhibition of Hmg1 with Lovastatin reverses the changes in steryl esters as expected. Does this treatment also affect Hmg1 NVJ accumulation and/or Nvj1 distribution?

We did not comment on this in the manuscript, but this treatment does not affect Hmg1 recruitment to the NVJ.

5. All the measurements of mevalonate pathway intermediates and products were performed in cells expressing Hmg2 and the levels of the protein under the conditions tested is unknown. Is it possible that Nvj1 affects the levels of Hmg2? To avoid this complications, the same measurements should have been done in hmg2D cells. Also the data shown in Figure 5 and 6 does not show that the effect is specific to AGR

Thank you. We have now done controlled mevalonate pathway flux experiments in a background with only Hmg1-GFP under a controlled ADH promoter (Figure 6J-M). We have also done these experiments in a mutant Nvj1 RK-28,29-AA that does not have Hmg1 (only Hmg1 here), and an intact NVJ is present. Here, there are still defects in mevalonate pathway flux, and Hmg1-mRuby3 protein levels are the same in Nvj1 WT and Nvj1 RK-28,29-AA backgrounds (Figure 6A-D; Figure 6figure supplement 1C). This strongly suggests that defects in mevalonate pathway flux and not due to Hmg1 protein levels, nor from Hmg2 alterations.

6. The authors propose that steryl esters are necessary for optimal AGR response. However, this possibility was not formally tested by preventing steryl ester formation, which can be easily done by deleting Are1 and Are2 acyl transferases.

As discussed above, we have modified the text to suggest that defects in AGR response my be due to defects in mevalonate production, which is necessary for many aspects of lipid and mitochondrial metabolism. In line with this, addition of mevalonate rescues NVJ1-ko associated growth defects following AGR (Figure 5H). Additionally, we now show that the Nvj1 RK-28,29-AA, which has an NVJ but cannot recruit Hmg1 to it, and has defects in mevalonate pathway flux, also has growth resumption defects (Figure 6E,F).

7. In Figure 6I, why does Hmg1-DsRed2 responds to AGR even in nvj1D cells. Presumably the tetramerization of DsRed is independent of glucose levels.

We do not completely understand this, but it is possible that the organization of the nuclear envelope and/or cytoplasm changes during AGR. It is also known that the fluidity of the yeast cytoplasm changes during glucose starvation, which may affect general macromolecular mobility, and could induce clustering of high molecular weight complexes (Joyner, e*Life*, 2016; https://elifesciences.org/articles/09376). This is part of an ongoing investigation.

8. The data on LCL^-^LDs is very interesting but it does not seem to belong to this story.

Thank you. We have removed this section and put in another ongoing project.

9. Seo and colleagues (eLife 2016) showed that AGR triggers LD autophagy. While the two phenotypes observed are not incompatible, the apparent differences of AGR in LD formation and/or consumption should at least be discussed.

Thank you. We now discuss the Seo *et al.* study and its relationship to our work in the Discussion.